# Benefits and Limitations of Communication in Multi-Agent Reasoning

**Michael Rizvi-Martel**[1]* **Satwik Bhattamishra**[2] **Neil Rathi**[3]

**Guillaume Rabusseau**[1] **Michael Hahn**[4]

[1]Mila & Université de Montréal    [2]University of Oxford    [3]Stanford University
[4]Saarland University

## Abstract

Chain-of-thought prompting has popularized step-by-step reasoning in large language models, yet model performance still degrades as problem complexity and context length grow. By decomposing difficult tasks with long contexts into shorter, manageable ones, recent multi-agent paradigms offer a promising near-term solution to this problem. However, the fundamental capacities of such systems are poorly understood. In this work, we propose a theoretical framework to analyze the expressivity of multi-agent systems. We apply our framework to three algorithmic families: state tracking, recall, and $k$-hop reasoning. We derive bounds on (i) the number of agents required to solve the task exactly, (ii) the quantity and structure of inter-agent communication, and (iii) the achievable speedups as problem size and context scale. Our results identify regimes where communication is provably beneficial, delineate tradeoffs between agent count and bandwidth, and expose intrinsic limitations when either resource is constrained. We complement our theoretical analysis with a set of experiments on pretrained LLMs using controlled synthetic benchmarks. Empirical outcomes confirm the tradeoffs between key quantities predicted by our theory. Collectively, our analysis offers principled guidance for designing scalable multi-agent reasoning systems.

## 1 Introduction

Chain-of-thought (CoT) prompting has become the de facto standard for tackling complex reasoning problems. By encouraging models to "think step-by-step", CoT significantly improves performance on tasks requiring mathematical and logical reasoning (Wei et al., 2022). Building on this, recent approaches view reasoning as a structured traversal over thoughts, exploring methods such as self-consistency (Wang et al., 2022), tree-of-thoughts (Yao et al., 2023), and stream-of-search (Gandhi et al., 2024). In parallel, post-training for large reasoning models (LRMs) increasingly relies on reinforcement learning over generated CoTs (OpenAI, 2025; Guo et al., 2025).

Despite these advances, several limitations have emerged. The reasoning abilities of LRMs degrade as the complexity of problem instances increases or as the context length grows (Shojaee et al., 2025; Sun et al., 2025). To address this, new approaches based on multi-agent collaboration (e.g., Zhang et al., 2024b; Tran et al., 2025; Xiao et al., 2025; Hsu et al., 2025) decompose complex tasks into simpler subproblems, coordinating multiple agents to achieve stronger performance. These frameworks offer promising near-term solutions, yet the theoretical underpinnings of their expressive capacity remain poorly understood. While the expressive power of Transformers with CoT prompting has been studied in depth (Merrill & Sabharwal, 2023; Amiri et al., 2025), little is known about the fundamental limits and tradeoffs of communication and resource allocation in multi-agent reasoning schemes. This gap motivates the central question of our work:

*From an algorithmic perspective, are there tasks that provably benefit from communication and dynamic resource allocation in multi-agent reasoning systems?*

We address this question by proposing a theoretical framework for analyzing the expressivity of collaborative multi-agent reasoning strategies. Our analysis applies to settings where both problem

---

*Corresponding author. Contact: `michael.rizvi-martel@mila.quebec`

complexity and context size scale, and focuses on three representative algorithmic families: state tracking, recall, and $k$-hop reasoning. For each task family, we establish bounds on the number of agents and the quantity of communication required, and we characterize the tradeoffs between these quantities. Finally, we complement our theoretical results with empirical validation using pretrained large language models. Our contributions are as follows:

- We propose a formalization of multi-agent reasoning systems grounded in the rich literature on Transformer expressivity.
- For three distinct families of algorithmic tasks—recall, state tracking and $k$-hop reasoning—we derive bounds on the number of agents and the communication required, highlighting the tradeoffs between these resources. These tasks capture key aspects of practical reasoning problems, making the results broadly applicable.
- We provide empirical validation of our theoretical insights by implementing the optimal communication protocols given by theory. Our analysis shows the performance in terms of accuracy, communication and token usage closely aligns with theoretical predictions.

Our work focuses on Transformer-based multi-agent systems which partition an input of size $N$ equally between $w$ agents, an abstraction of many settings where multiple agents cooperate by taking responsibility for different parts of the input, such as different document chunks in long-context summarization or question answering, corpus shards or knowledge-graph subgraphs in multi-agent RAG, web pages or site sections in browser-style agents, and map–reduce pipelines where workers process disjoint partitions before a coordinator aggregates the partial results (e.g., Zhou et al., 2025; Chang et al., 2025; Zhang et al., 2024a; Guo et al., 2024; Salve et al., 2024; Yang et al., 2025b; Xiao et al., 2025; Liu et al., 2025; Xu et al., 2025).

Our results reveal three distinct regimes for multi-agent tasks, each instantiated by natural tasks of broad relevance (Table 1). First, some tasks require almost no communication overhead even when the input is partitioned between agents, such as key-query retrieval. Second, some tasks not only allow partitioning but also benefit from it, achieving reduced wall-clock time compared to a single-agent setup; state tracking is a prime example. Finally, some tasks can be solved through partitioning but require significant communication among agents, such as reasoning over multiple hops.

|                   | Size            | Depth                               | Communication           |
| ----------------- | --------------- | ----------------------------------- | ----------------------- |
| Associative recall | $\Theta(w)$     | $\Theta(1)$                         | $\Theta(1)$             |
| State tracking    | $\Theta(N)$     | $\mathcal{O}(\frac{N}{w} + \log w)$ | $\Theta(w)$             |
| $k$-hop reasoning | $\mathcal{O}(wk)$ | $\mathcal{O}(k)$                   | $\Theta(k)$ (when $w > 1$) |

Table 1: Summary of results. $w$ denotes the number of agents. $N$ represents the length of the input. *Size* corresponds to total computation. *Depth* loosely corresponds to wall-clock time. *Communication* refers to the overall amount of communication between agents. We will define these formally in Section 3. $\mathcal{O}(\cdot)$ indicates existence of a protocol; $\Theta(\cdot)$ indicates that we prove it optimal.

## 2 MODEL OF TRANSFORMERS

We assume causally masked (decoder-only) unique hard attention Transformers (UHAT) (e.g., Hahn, 2020; Hao et al., 2022; Yang et al., 2024a; Amiri et al., 2025; Jerad et al., 2025a; Bergsträßer et al., 2024), a popular abstraction where attention heads concentrate attention on the position maximizing the attention score. Some work suggest that pretrained models concentrate their attention on only a few positions (Voita et al., 2019; Clark et al., 2019). Importantly, UHAT subsumes expressivity of ordinary softmax Transformers with fixed precision (Jerad et al., 2025a), making it a plausible model of Transformers in the regime of long contexts and large reasoning problems (see Appendix I).

Each layer of a Transformer has an attention block followed by an MLP block. The attention block takes as input $\mathbf{X} \in \mathbb{R}^{N \times d}$ and applies the operation $\text{Att}(\mathbf{X}) = f^{\text{Att}}(\mathbf{X}\mathbf{W}_Q\mathbf{W}_K^\top\mathbf{X}^\top)\mathbf{X}\mathbf{W}_V^\top$ where $\mathbf{W}_Q, \mathbf{W}_K, \mathbf{W}_V \in \mathbb{R}^{d \times m}$ and $f^{\text{Att}}(\cdot) = \text{UHAT}(\cdot)$, where for any matrix $\mathbf{A} \in \mathbb{R}^{N \times M}$:

$$\text{UHAT}(\mathbf{A})_{i,j} = \begin{cases} 1 & \text{if } j = \arg\max \mathbf{A}_{i,:} \\ 0 & \text{else} \end{cases}, \tag{1}$$

where in case of a tie, the rightmost element is selected. Multi-head attention with $H$ heads is defined as M-Att$_H(\mathbf{X}) = [\text{Att}_1(\mathbf{X}), \dots, \text{Att}_H(\mathbf{X})]\mathbf{W}_O$ where each Att$_i(\mathbf{X})$ has its own set of parameters. The matrix $\mathbf{W}_O \in \mathbb{R}^{mH \times d}$ projects the concatenated vector to a vector of dimension $d$. For an input $\mathbf{X} \in \mathbb{R}^{N \times d}$, the output of a Transformer layer is $\psi(\text{M-Att}_H(\mathbf{X})) \in \mathbb{R}^{N \times d}$ where $\psi : \mathbb{R}^d \to \mathbb{R}^d$ corresponds to the function computed by the MLP. The model has access to arbitrary positional embedding vectors $p_i \in \mathbb{R}^d$, for each $i \in [N_{max}]$, where $N_{max}$ is the model's context window and $[N_{max}]$ denotes $\{1, \dots, N_{max}\}$.

## 3 Formalization of Multi-Agent Systems

We formalize multi-agent systems as graphs, with a node representing an agent at a given timestep, and edges describing both the emission of CoT tokens, and communication between different agents. We discuss connections to applied systems in Section 3.1, and illustrate the definition in Figure 1.

**Definition 3.1** (Multi-agent system). Let $\Sigma$ be a (finite or infinite) input alphabet and $\Xi \supset \Sigma$ an (infinite) CoT alphabet. For broad generality, depending on the task, we're allowing both input and CoT alphabets to grow with the input length, such that $\Sigma_1 \subset \Sigma_2 \subset \cdots \subset \Sigma$ and $\Xi_1 \subset \Xi_2 \subset \cdots \subset \Xi$, where $|\Sigma_N|, |\Xi_N| \in \mathcal{O}(\text{poly}(N))$. We write the set of input strings as $\mathcal{S} := \bigcup_{N=1}^{\infty} (\Sigma_N)^N$. We reserve agent identifiers $\text{ID}_1, \dots, \text{ID}_N \in \Xi_N$.

A *multi-agent system* $\mathcal{A}$ maps strings $x \in \mathcal{S}$ to labeled DAGs $\mathcal{A}(x)$ with $w(x) \leq |x|$ agents where:

1. Each node is uniquely labeled as $T_i^{(t)}$, where $i \in [w(x)]$ and $t \in \mathbb{N}$. Informally, it represents agent $i$'s state at time $t$. For each agent $i \in [w(x)]$, there is $D_i \in \mathbb{N}$ such that there are nodes $T_i^{(t)}$ exactly for $t \in [D_i]$ and for no other $t$.
2. We define two types of edges:
   (a) *communication edges* $\{c, \sigma\}$ ($\sigma \in \Xi_{|x|}$) from $T_i^{(t)}$ to $T_j^{(t+1)}$, which represent communicating a symbol between two different agents ($i \neq j$)
   (b) *CoT edges* $\{a, \sigma\}$ ($\sigma \in \Xi_{|x|}$), which correspond to autoregressive decoding of the model from $T_i^{(t)}$ to $T_i^{(t+1)}$
3. Every node $T_i^{(t)}$ ($t > 1$) has exactly one incoming edge.
4. Every node $T_i^{(t)}$ can have (i) no outgoing edge, (ii) one outgoing edge, (ii) outgoing edges edge with the same label $\{c, \sigma\}$ to each other agent, $j \neq i$.
5. One agent $i \in [w(x)]$ is designated as a *manager agent*.

By definition, agents can only send or receive a single token $w \in \Xi$ at a time.[*] Every agent can only receive one incoming edge at a time. Intuitively, the symbol provided by the incoming edge at time $t$ (whether it is a CoT or communication edge) is appended to the agent's context at this time step.

**Definition 3.2** (Complexity of Multi-Agent System). We use the following notions to characterize the complexity of a multi-agent system:

- *Computation depth* is the length of the longest path in the graph, irrespective of edge type. We write $Depth(N)$ for the maximum depth on any input $x$ of size $|x| \leq N$. Computation depth is a proxy for the wall-clock time.
- *Width* of the graph corresponds to the number of agents in the system. Typically we use $w(N)$ when the number of agents is a function of input length. As the input is partitioned into chunks of size $N/w$, we consider $w(N) \in [N]$.
- *Size* corresponds to the number of nodes in the graph. We write $Size(N)$ for the maximum size on any input of size $N$.
- *Communication budget* is the number of nodes with an outgoing communication edge.

We say a multi-agent system $\mathcal{A}$ *computes* a function $f : \mathcal{S} \to \Sigma$ if for all $x \in \mathcal{S}$, the last CoT edge of the manager agent in $\mathcal{A}(x)$ has the label $f(x) \in \Sigma$.

---

[*]An extension to words of bounded length $w \in \Xi^{\leq C}$ would be straightforward, but we find it easiest to formalize the setup with single-token messages. Our theoretical results hold irrespective of this choice (App. G).

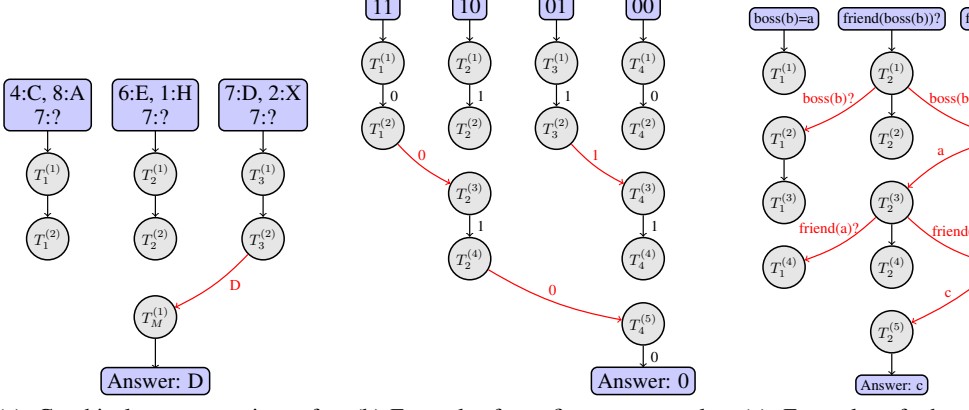

(a) Graphical representation of recall protocol (Sec. 4.2). $T_1, T_2$ and $T_3$ are worker agents given chunks of 2 key-value pairs. Only $T_3$ has the query in its context and thus communicates the answer to the manager $T_M$ after reasoning.

(b) Example of a prefix sum protocol for state tracking (Sec. 4.3) on input 11100100. Here $T_2$ and $T_4$ act as intermediate managers composing together the answers of $T_1$ and $T_3$ with their own. $T_4$ also acts as the final manager, providing the final output.

(c) Example of the Iterative Query protocol (Sec. 4.4). Each agent holds one fact. The full query, friend(boss(b)), is managed by $T_2$, which receives answers at $t = 3, 5$ and broadcasts followup queries at $t = 2, 4$.

Figure 1: Graphical representations of the protocols analyzed in Section 4.

**Definition 3.3** (Agent computation). Given a multi-agent system $\mathcal{A}(x)$ on input $x \in \mathcal{S}$, for each agent $i \in [w(x)]$, we construct a string $\xi(i) \in (\Xi_{|x|})^*$. Intuitively, this is the sequence of tokens that is processed by this agent over the course of reasoning. The string $\xi(i)$ is constructed as follows:

1. First, take the input chunk $x_{\left\lceil |x| \cdot \frac{i}{w} \right\rceil, \cdots, \left\lceil |x| \cdot \frac{i+1}{w-1} \right\rceil} \in \mathcal{S}$
2. Then append $\text{ID}_i \in \Xi_{|x|}$
3. Then traverse the nodes associated to the agent, $T_1^{(i)}, T_2^{(i)}, \ldots$:
   (a) If there is an incoming communication edge $\{c, \sigma\}$, append the token $\text{RECEIVE}\_\sigma$.
   (b) If there is an outgoing CoT edge $\{a, \sigma\}$, append the symbol $\sigma$.
   (c) If there is an outgoing communication edge,
      i. Either the message is sent to a single agent $j$ – in this case, append the token $\text{SEND}\_\sigma\_\text{ID}_j$ – or,
      ii. the message is broadcast to all agents, append the token $\text{BROADCAST}\_\sigma$.
4. We append the EOS symbol.

We assume all tokens to be part of $\Xi_{|x|}$.[†] A Transformer $T$ *implements* $\mathcal{A}(x)$ on input $x \in \mathcal{S}$ if and only if each of these strings $\xi(i)$ fits into the Transformer's context size, and the transformer predicts all tokens arising from outgoing edges and EOS (cases 3b,3c,4) when run autoregressively on $\xi(i)$. Intuitively, in each reasoning step, the transformer generates the next token, unless it is overridden by incoming communication.

A protocol $\mathcal{A}$ is *expressible* in UHAT if, for each input length $n$, there is a UHAT Transformer $T_n$ implementing $\mathcal{A}(x)$ on each input $x$ with length $|x| \leq n$, and the number of heads and layers are uniformly bounded across all Transformers $T_n$. We do not require the width $d$ to stay bounded; e.g., growing width can allow positional encodings to keep unboundedly many positions distinct.

The above generalizes the setup of Amiri et al. (2025) to the multi-agent setup. Requiring the system to be implemented by models with bounded layers and heads across input lengths is a very weak assumption, much weaker than the uniformity of the Transformer across input lengths often required in theoretical work on CoT (e.g., Pérez et al., 2019; Merrill & Sabharwal, 2023) – nonetheless, it allows us to prove essentially matching upper and lower bounds on the cost of multi-agent systems.

---

[†]We assume single tokens for simplicity; they could be bounded-length words without change to our results.

### 3.1 CONNECTION TO APPLIED WORKS

Our formalization covers a broad range of LLM-based protocols which (i) split long contexts into non-overlapping chunks, (ii) process these chunks in parallel with worker agents, (iii) relay info to a manager to generate the final answer. The primary distinctions lie in coordination: CoA (Zhang et al., 2024b), LLM×MapReduce (Zhou et al., 2025), NexusSum (Kim & Kim, 2025), AgentSimp (Fang et al., 2025), and Multi$^2$ (Cao et al., 2025) run workers in parallel with minimal communication, whereas LongAgent (Zhao et al., 2024), XpandA (Xiao et al., 2025), AgentGroupChat-V2 (Gu et al., 2025), and certain task-specific pipelines (e.g., DocAgent (Yang et al., 2025a), Multi-Agent QG (Wang et al., 2025a)) support targeted message-based communication to resolve conflicts. Each of these approaches can be described by a communication graph as in Figure 1. Notably, all of these implement communication using messages written to the recipient agent's context, consistent with our framework.

## 4 RESULTS

### 4.1 GENERAL RESULTS: THREE REGIMES FOR DEPTH AND COMMUNICATION

In this section, we present theoretical results which hold for all tasks and all multi-agent systems which follow from Definition 3.1. Throughout, by "multi-agent systems", we always refer to systems computed by Transformers. The first result we present relates to the *size* of the system:

**Proposition 4.1** (Conservation of size)**.** *Any protocol can be converted into an equivalent single-agent protocol with the same size up to constant factor.*

The proof idea is to construct a single agent that alternates between simulating each of the agents of the original protocol, see Appendix C.2. Thus, the size of the protocol cannot be reduced using multiple agents beyond a constant factor. However, we will see that it may increase in some tasks with the number of agents. In a multi-agent protocol, two other key determinants of cost are (i) depth, and (ii) communication budget. There are two fundamental *a priori* considerations about these quantities. First, any protocol as defined in Definition 3.1 satisfies the inequality:

$$\frac{Size(N)}{w(N)} \le Depth(N) \tag{2}$$

*Proof.* The size is upper-bounded by the number of agents times the maximum number of nodes assigned to any individual agent, which is upper-bounded by the maximum length of any path. □

This inequality might lead one to hope that multi-agent protocols reduce depth even if size cannot be reduced. We show that such a gain in depth is realizable in some tasks (Section 4.3), but there are other tasks where *no asymptotic gain in depth is possible* (Section 4.4). A second fundamental observation is that reduction in depth due to multi-agent reasoning is only possible at an increase in communication. In fact, any task solvable at bounded communication cost can already be solved at bounded depth by a single agent, ruling out gains in depth from a multi-agent setup:

**Proposition 4.2.** *Consider a task with a multi-agent system whose communication budget is $\mathcal{O}(1)$ in $N$ across all $w \in [N]$. Then this task has a single-agent CoT with depth (and hence size) $\mathcal{O}(1)$.*

See proof in Appendix C.2. Taken together, this leaves us with *three distinct feasible regimes* of multi-agent reasoning (Table 2). The first one (Section 4.2) is where both depth and communication are $\mathcal{O}(1)$ as the number of agents increases. In this setting, multi-agent setups simply enable processing larger contexts, *without associated cost*. The second regime (Section 4.3) is where depth can be reduced by using more agents (almost up to (2)), though at the cost of increased communication. That is, there is a *depth-communication tradeoff*. The third regime (Section 4.4) is where, at

| | **Communication** | |
|---|---|---|
| **Depth** | $\mathcal{O}(1)$ | *increases* |
| $\sim$Size/$w$ | impossible | Section 4.3 |
| *No Gain* | Section 4.2 | Section 4.4 |

Figure 2: Three possible and one impossible regimes for depth-communication tradeoffs.

least in the worst case, multi-agent setups require a large amount of communication, without reduction to the required depth. In this regime, multi-agent setups allow processing larger contexts, with a *high cost of communication and no reduction in depth*. A seeming fourth regime, where communication stays $\mathcal{O}(1)$ but depth decreases as (2), is impossible by Proposition 4.2. Importantly, we will next demonstrate that all three regimes are instantiated by naturalistic tasks.

## 4.2 ASSOCIATIVE RECALL

The first task we consider is simple, associative recall. In this setup, given multiple key value pairs, and a queried key, agents must return the associated value. In this case, multi-agent setups permit processing a larger input without associated cost in communication or depth:

**Proposition 4.3.** *Given an input consisting of $N$ pairs $(x_i, y_i) \in \Sigma_N \times \Sigma_N$, and a query $x$, consider the task of retrieving the (unique) $y$ such that $(x, y)$ appears in the input. Assume that the input is partitioned disjointly into parts provided to $k$ agents, which also have access to the query. Then they can solve the task with depth $\mathcal{O}(1)$ and communication $\mathcal{O}(1)$.*

*Sketch of proof (Full proof in Appendix C.3).*     Each agent uses attention to check if the query $x$ appears in the input, and uses an induction head to retrieve the associated $y$ if it appears. By design, only one agent will find such a $y$; it then reports it to a designated manager agent that outputs $y$.  $\square$

> **Tradeoffs for Simple Retrieval**
> 1. Computation depth $\mathcal{O}(1)$
> 2. Number of agents $w(N)$ and chunk size: $\frac{N}{w(N)}$
> 3. Communication budget $\mathcal{O}(1)$
> 4. Size: $\mathcal{O}(w(N))$
>
> is both realizable and optimal for retrieval.

## 4.3 STATE TRACKING

Another foundational task is state tracking. State tracking is at the heart of many reasoning problems, such as tracking chess moves in source-target notation, evaluating Python code, or entity tracking (Kim & Schuster, 2023; Merrill et al., 2024). State tracking can be conveniently formalized in terms of evaluation over finite monoids (e.g., Merrill et al., 2024; Grazzi et al., 2025):

**Definition 4.1** (State tracking problem). Let $M$ be a finite set, and $(M, \cdot)$ a finite monoid ($M$ with an identity element and associativity). A state tracking problem on $M$ is defined as sending a sequence $m_0 m_1 \ldots m_k \in M^*$ to $m_0 \cdot m_1 \cdot \ldots \cdot m_k \in M$. Here, the input alphabet is $\Sigma := M$.

Elements of the monoid represent operations (e.g., list manipulation instructions in Python or chess moves). Composing them leads to new monoid elements (e.g., compositions of instructions, or a sequence of chess moves). This problem class subsumes deciding membership for all regular languages, such as PARITY, which corresponds to the monoid $(\{0, 1\}, \oplus)$. Amiri et al. (2025) showed that PARITY requires a CoT of length $\Omega(N)$. Can a multi-agent system with a large amount of total communication do better? In terms of the *size* of the graph, this cannot be the case:

**Proposition 4.4.** *Any multi-agent system computing* PARITY *requires size $\Omega(N)$.*

The proof is in Appendix C.4. However, if we consider a parallel computation budget, we can obtain a speedup in the *depth* of the computation graph. We assume the setup where each agent receives a disjoint contiguous substring of the input. Then:

**Proposition 4.5.** *Let $M$ be a finite monoid. There exists a communication protocol with $w(N) = N$, depth $\mathcal{O}(\log N)$ computing the state tracking for $M$.*

The key idea for this protocol is to compute the prefix sum (or recursive parallel scan) algorithm with the LLM agents, as shown in Figure 1(c). The above protocol has a width of $N$ agents, but it can be generalized to other widths given by some function $w(N)$ of the input size $N$:

**Proposition 4.6.** *Given a finite monoid $M$ and any number of agents ($w : \mathbb{N} \to \mathbb{N}$ with $w(N) \in [N]$), there exists a $\mathcal{O}(\log w(N) + \frac{N}{w(N)})$ depth and $\mathcal{O}(N)$ size multi-agent system computing state tracking on $M$ with communication budget $w(N)$.*

This means that given enough parallel computation budget, we indeed recover a speedup in terms of effective or wall-clock time. The proof for this result is given in Appendix C.4; Proposition 4.5 is a corollary. The above result is essentially optimal, in that essentially no shorter depth is attainable:

**Proposition 4.7** (Optimality, see App. C.4 for proof). *Assume the finite monoid $M$ is a nontrivial group, and $\mathcal{A}$ a multi-agent system computing state tracking over $M$. Then $\mathcal{A}$ has $\Omega(w(N))$ communication budget, and computation depth $\Omega(\frac{N}{w(N)})$.*

We summarize our results for state tracking below:

**Tradeoffs for State Tracking** Assume $w(N) \in [N]$ agents, each provided a disjoint contiguous portion of the input. Then

1. Computation depth $\mathcal{O}\left(\log w(N) + \frac{N}{w(N)}\right)$
2. Number of agents: $w(N)$ and chunk size: $\frac{N}{w(N)}$
3. Communication budget $\mathcal{O}(w(N))$
4. Size: $N$

are realizable for performing state tracking. Communication budget and size are optimal. Computation depth is optimal at least up to $\mathcal{O}(\log w(N))$.

### 4.4 MULTI-HOP REASONING

We instantiate the third regime with $k$-hop reasoning (e.g., Yang et al., 2024b; Wang et al., 2025b; Yao et al., 2025). In this task, we have a domain $\mathcal{D}$ of objects and a vocabulary $\mathcal{F}$, intended to denote functions. We have a set of $N$ facts $f(x) = y$ ($f, x, y \in \Sigma_N$) contextually given, where for each $x$ and $f$ at most one such fact is included. Each agent receives a disjoint equal sized partition of the set of facts, and a common query of the form $f_1(\dots(f_k(x))\dots)$ where $f_i \in \mathcal{F}$, $x \in \mathcal{D}$. The overall size of the input is $N + k$; each agent has $\frac{N}{w}$ facts and the $k$-hop query.

**Proposition 4.8.** *Let the number of agents be $w : \mathbb{N} \to \mathbb{N}$ ($w(N) \in [N]$). The $k$-hop composition task with $N$ facts can be solved with computation depth $\mathcal{O}(k)$, communication budget $\mathcal{O}(k)$, and size $\mathcal{O}(w(N) \cdot k)$. The communication budget is optimal. The computation depth $\mathcal{O}(k)$ is optimal at least up to a $\log(N + k)$ factor.*

The proof is in Section C.5. The idea is that worker agents perform an iterative lookup, where each agent tries to find the next answer in the own context, with one step for each of the $k$ hops $f_k(x), f_{k-1}(f_k(x)), \dots, f_1(\dots(f_k(x))\dots)$. The regime of this task is different from the previous ones; in the worst case, there is no reduction of computation depth when increasing the number of agents: Depending on how the facts are distributed among the agents, computation depth and communication budget may be $\Omega(k)$ in the worst case, as long as more than one agent are involved. The intuition here is that the relevant facts can be distributed between different agents, making iterative lookup the optimal strategy. We thus have:

**Tradeoffs for $k$-hop Composition** for $k$-hop composition and $N$ facts, when $w(k) > 1$:

1. Computation depth $\mathcal{O}(k)$
2. Number of agents: $w(k)$ and chunk size: $\frac{N}{w(k)}$
3. Communication budget $\mathcal{O}(k)$
4. Size: $\mathcal{O}(wk)$

are realizable for $k$-hop composition. Communication budget is optimal. Computation depth is optimal at least up to a $\log(N + k)$ factor.

## 5 EXPERIMENTAL VALIDATION

In this section, we experimentally validate whether the protocols of Section 4 work in practice and if computation depth and communication exhibit the three predicted regimes. We evaluate Llama-3.3-70B-Instruct-Turbo and Llama-3.1-8B-Instruct-Turbo (results in Appendix F) on associative recall, state tracking and $k$-hop reasoning tasks in order to empirically validate each of the three regimes

analyzed in theory. We employ pretrained LLMs which are prompted with their roles in the protocol and the instructions to solve the task. We use hard coded communication protocols similar to the protocol implementation of Zhang et al. (2024b). For more details please refer to Section E.

## 5.1 RECALL

We validate experimentally the abilities of different multi-agent systems to perform associative recall using the needle-in-a-haystack test (Kamradt, 2024). This task involves finding a "needle" (an answer to some query) in a large document (the "haystack"). We use self-consistency with majority voting (Wang et al., 2022) as our baseline and use an implementation of Chain-of-Agent (CoA) for the optimal protocol, given its similarity.

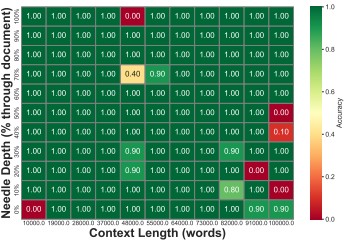
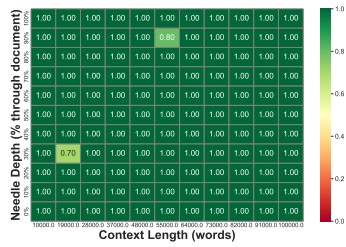
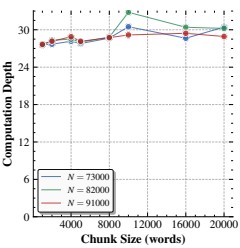

(a) Heatmap of accuracy vs needle depths and context lengths for Majority Voting. Performance degrades for large contexts.

(b) Heatmap of accuracy vs needle depths and context lengths for CoA. Performance remains constant across context size.

(c) Token usage for CoA. Usage remains constant across chunk size and context length, which is consequent with our theory.

Figure 3: Results for the needle-in-a-haystack test.

**Needle-in-a-haystack**  Figure 3 shows the performance of both Majority Voting and CoA across needle depth (how far in the corpus the needle is) as well as context length. As it can be seen across Figure 3(a), Majority Voting degrades in performance as sequence length increases. We note also that extremities are also slightly problematic; Majority Voting also fails when needle is at the beginning or end of the corpus. CoA does not suffer from such degradation: the "divide-and-conquer" strategy of the protocol makes the task more manageable.

## 5.2 STATE TRACKING

We next experimentally evaluate multi-agent systems on state tracking tasks. We evaluate models on PARITY i.e., determining if the number of 1s in a bitstring is even or odd.

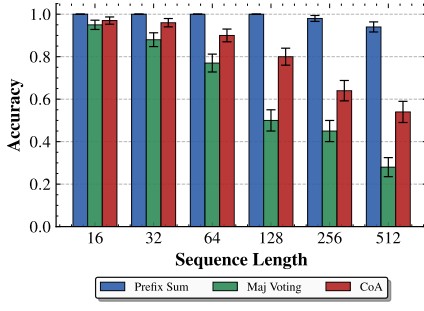
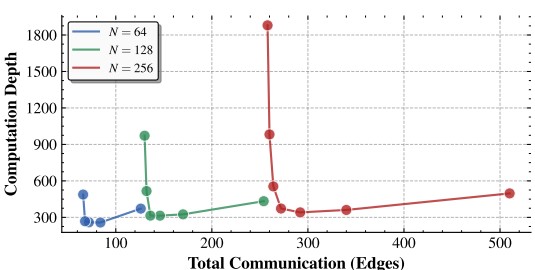

(a) Llama-70B accuracy on PARITY for different sequence lengths. Prefix Sum represents the theoretically optimal communication protocol.

(b) Computation depth against the total amount of communication used. This trend is consistent with the $N/w(N)$ computation depth vs $w(N)$ total communication tradeoff predicted in Section 4.3.

Figure 4: Empirical validation for PARITY.

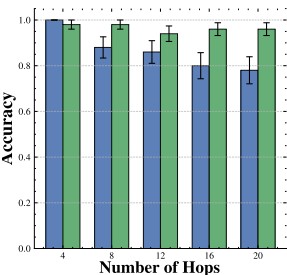
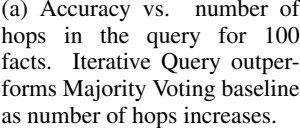
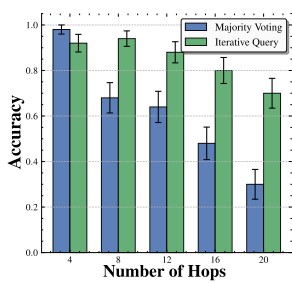
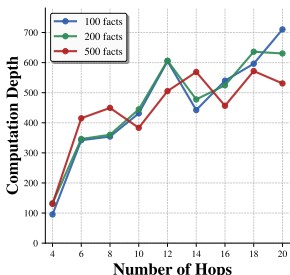

(a) Accuracy vs. number of hops in the query for 100 facts. Iterative Query outperforms Majority Voting baseline as number of hops increases.

(b) Accuracy vs number of hops in the query for 500 facts. The difference in performance is more pronounced in this regime.

(c) Computation depth vs. number of hops in the query. Computation depth shows an increasing trend as hop count increases.

Figure 5: Empirical validation for $k$-hop reasoning.

Figure 4(a) shows that Prefix Sum consistently outperforms all other methods, especially as sequence length grows. Compared to Majority Voting, CoA degrades less with longer sequences, supporting our intuition that chunking complex reasoning into shorter parts helps. In terms of communication, Figure 4(b) shows the tradeoff between the computation *depth* and the total amount of communication for Prefix Sum, calculated by summing the average token usage at every level. This trend is consequent with the theoretically predicted tradeoffs between communication and computation. Indeed, in Section 4.3 we predict a tradeoff between depth $N/w(N)$ and total communication $w(N)$. We note, however, a slight increase in computational depth for high levels of communication. This is due to poor instruction following; models add a constant token overhead by repeating the query and explaining the procedure, especially noticeable in high-communication regimes.

## 5.3 $k$-HOP REASONING

Finally, we evaluate models on a $k$-hop reasoning task, where agents are given *facts* (e.g., "Paula is the boss of Mary") and a *query* (e.g., "Who is the boss of the friend of George?"). Task difficulty depends on the number of facts and query hops. We compare two protocols: Majority Voting and Iterative Query, an implementation of the protocol proved optimal in Section 4.4. As we can see in Figure 5, Iterative Query generally outperforms Majority Voting. This trend is accentuated as the number of hops increases. We note that for the smallest number of hops (four), there are cases where Majority Voting outperforms Iterative Query; we posit that the probability of failing at a given round seems to outweigh the difficulty of retrieving facts in a larger corpus in this regime. Finally, we analyze how the computation depth varies as a function of number of hops in the query. As can be seen in Figure 5(c), the trend we observe is consistent with theory; as depth of queries increases, so does the computation depth.

## 6 DISCUSSION AND CONCLUSION

In summary, our work provides principled foundations for understanding the algorithmic benefits and limitations of collaborative multi-agent systems. By formalizing communication and resource tradeoffs, we bridge theoretical analysis with empirical observations, shedding light on when collaboration enhances reasoning efficiency and when it imposes inherent costs. These results open new avenues for designing reasoning systems that balance scalability, expressivity, and performance..

**Relationship to Self-Consistency** Our results show that multi-agent systems with sophisticated communication protocols outperform majority-vote strategies; Mirtaheri et al. (2025) find that self-consistency with polynomially many agents yields limited gains on tasks hard for Transformers. In Appendix H, we show a separation result between an optimal protocol and majority voting strategies. We give a short rundown of this result here. Informally, we define a majority voting strategy using a set of $w(N)$ agents, each of which is run on the full input and terminates with a single-token

response; the overall system returns the most common response. Using this definition, we show a *super-polynomial* separation between majority voting and PrefixSum:

**Proposition 6.1** (Informal). *Consider a majority voting scheme with $w(N)$ agents computing* PAR-ITY *over length-$N$ inputs with a CoT length (or computation depth) of $\mathcal{O}(\log N)$. This scheme must have $w(N) = 2^{\Omega(N^c)}$ agents for some constant $c > 0$.*

In contrast, PrefixSum attains perfect accuracy with computation depth $\mathcal{O}(\log N)$ and $w(N) = N$. The proof of this statement as well as an in-depth discussion of its implications can be found in Appendix H.

**Implications for Protocol Design** Our results characterize depth (wall-clock time) and communication cost, identifying three regimes of multi-agent systems (Figure 2) with implications for multi-agent LLM design. Systems with many workers and a single manager (e.g., CoA Zhang et al. (2024b), LLM×MapReduce (Zhou et al., 2025), NexusSum (Kim & Kim, 2025), AgentSimp (Fang et al., 2025), Multi$^2$ (Cao et al., 2025)) only shift the context bottleneck to the manager, risking errors when aggregating many responses. To address this, we propose a prefix-sum–style cascade: iterative summarization reduces the final-agent bottleneck, with branching factor and depth as tunable hyperparameters. We also believe the Iterative Query protocol for $k$-hop reasoning could have practical relevance. For complex queries, a similar architecture may be promising: a manager first decomposes the main query into subqueries, each processed through iterative worker–manager communication rounds, with the manager updating the query after every round

**Relation to Parallel Computing Frameworks.** Our setting is related to classic models of parallel and cooperative computation: communication complexity, PRAM (Fortune & Wyllie, 1978), Massively Parallel Computation (Im et al., 2023), BSP (Valiant, 1990), and LOCAL (Peleg, 2000). Conceptually, our measures of *Computation Depth* and *Size* mirror *Time* (parallel steps) and *Work* (total operations) in PRAM. The key difference is that our analysis relies on the expressivity of the Transformer architecture. This formalization leads to the sharp contrast between Associative Recall (Section 4.2), which a Transformer can perform in one attention step, and State Tracking (Section 4.3), which is more challenging and requires an explicit multi-step reasoning chain. Other agent choices yield different predictions: multi-agent systems instantiated with recurrent networks would make retrieval harder (Bhattamishra et al., 2024; Arora et al., 2023) and may enable more efficient state tracking than Transformers; unconstrained agents solve both tasks in constant time; and Turing machines or RAM models require linear time for both. Frameworks that treat agents as unconstrained processors (with memory/communication as the main bottleneck) predict low depth for both tasks, while frameworks that model agents as RAM-like units predict high depth. It is only by considering the capabilities of Transformers that we achieve a more fine-grained analysis appropriate to LLM-based multi-agent systems, predicting tradeoffs realized by actual LLMs.

**Limitations and Future Work** There are many directions in which this work could be extended. Empirically, ideas discussed in the above paragraph could be incorporated into the design of novel multi-agent systems. Theoretically, our work could be extended to different algorithmic tasks e.g., graph reachability or to different multi-agent paradigms such as adversarial games or cooperative reinforcement learning tasks, where agents collaborate to reach a common goal.

## REPRODUCIBILITY STATEMENT

We provide a complete reproducibility package to facilitate replication of our results. Code to reproduce all experiments can be found at `https://github.com/michaelrizvi/coa-algorithmic`. Appendix E details the experimental setup, including the hyperparameters and parameter ranges considered, as well as all system prompts used. Complete proofs for all propositions presented in the paper are included in Appendix C.2.

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

APPENDIX

## A  USE OF LARGE LANGUAGE MODELS

We used a large language model (GPT-5) in two ways during the research and writing of this paper. First, we used it to refine phrasing and to provide revision suggestions on draft manuscripts. Second, we used it to conduct preliminary literature reviews of the field. Research ideas, proof, and empirical analysis were conducted by the authors.

## B  CLARIFICATIONS

    **(i)**  *Similar work has already been done in the parallel computation and communication complexity literatures. What is the key difference here?*

Indeed, there are various frameworks for modeling parallel and cooperative computation (such as communication complexity, Parallel RAM (Fortune & Wyllie, 1978), Massively Parallel Computation (Im et al., 2023), BSP Valiant (1990), LOCAL (Peleg, 2000)). There are some conceptual similarities, for instance our analysis of Computation Depth and Size mirrors the concepts of Time (number of parallel steps) and Work (total operations across all processors) in the Parallel RAM model.

Our work differs in making essential use of arguments about the expressivity of the Transformer architecture, and no prior framework would have enabled us to show the set of results in this paper. For instance, the very different behavior between Associative Recall (Section 4.2) and State Tracking (Section 4.3) builds on properties of the Transformer architecture: A Transformer performs retrieval in one step with attention, whereas state tracking is challenging unless a full reasoning chain is used. In contrast, if one instantiates a multi-agent system based on recurrent networks, then retrieval would be challenging (Bhattamishra et al., 2024; Arora et al., 2023) and state-tracking could perhaps be done more efficiently than a Transformer-based system. An unconstrained agent performs both in constant time; a Turing machine or a RAM needs a linear number of steps in both cases. Models based on unconstrained individual agents (with memory and communication as the primary bottleneck, as in PRAM) predict low computation depth in both tasks; models where individual agents are based on RAM (as in PRAM) predict similarly high depth in both tasks. It is only by considering the specific abilities of Transformers that we achieve a more detailed analysis appropriate to LLM-based multi-agent systems, predicting tradeoffs realized by actual LLMs.

(ii) *What are the impacts/takeaways of the experimental results for practitioners?*

The main objective of our experiments is to validate the theoretical claims we provide. To this end, we provide experimental results which corroborate token usage with theoretical predictions about computation depth. Moreover, we report accuracy and compare to a self-consistency baseline in order to illustrate that using the optimal protocol for a given task can also lead to performance gains. We stress that the implementation of the protocols we use is kept simple. Our work is an analysis paper, not a methods paper and thus its main aim is to better understand the interplay of agent count, token usage and communication in multi-agent systems. Refinement and engineering of the considered protocols is left for future work

(iii) *What is the practical impact of the results? Do the results relate to any specific NLP tasks?*

RECALL, PARITY and $k$-hop reasoning are fundamental problems that serve as simple models of reasoning tasks and are of broad interest. RECALL has been shown by Arora et al. (2023) to play in important part in the impressive linguistic abilities of modern transformer-based LLMs. PARITY and other state tracking problems are of great interest as they are directly connected to reasoning problems such as code evaluation, entity tracking in linguistics and generally require some form of world modeling abilities (Merrill et al., 2024; Kim & Schuster, 2023; Rizvi-Martel et al., 2024). Moreover, PARITY is a prime example of a sensitive function, which have been shown to be difficult for Transformer models (Hahn, 2020; Hahn & Rofin, 2024). Finally, $k$-hop reasoning is foundational to many aspects of reasoning involving the composition of multiple reasoning steps.

(iv) *Why are there no experiments on large reasoning models such as OpenAI-o3 or DeepSeek-R1?*

In the experiments, we typically prompt models to use a specific reasoning strategy for consistency. LRMs typically reason in the way they see best, not necessarily respecting the reasoning strategy given in the prompt. Moreover, using the TogetherAI API, such models typically reason using a large CoT which is hidden to the user, thus rendering it impossible to monitor the exact CoT used by the model.

Moreover, in many experiments, we evaluate token usage as a proxy for certain metrics e.g. computation depth. The long reasoning chains produced by these models make it difficult to see any trends in the token usage as problem complexity increases.

(v) *Section 3 assumes that each agent can only receive bounded communication at each step. What about protocols where each agent can see each other agent's tokens, as in Group-Think (Hsu et al., 2025)?*

One of our key aims here is to understand the cost of communication introduced in multi-agent setups and how it trades off with computation depth, which is easiest to do if we consider a framework that explicitly counts each communication link. In contrast, this is not straightforward in scenarios where all agents have access to the same information.

It may be possible to extend the current framework to protocols with unbounded communication, but this would require careful consideration of what it would mean for transformers to access the contexts of all agents, and how to quantify the amount and cost of communication in scenarios where, at least in principle, all agents have access to the same information.

(**vi**) *The paper assumes that the input is partitioned between agents. What about frameworks such as multi-agent debates and voting, where multiple agents process the same input?*

Indeed, various papers have proposed strategies where different agents process the same input and solve the problem through mechanisms such as debating or voting (Wang et al., 2022; Dhuliawala et al., 2024; Du et al., 2023; Wang et al., 2024b). An important special case is Self-Consistency (Wang et al., 2022), where agents solve a problem independently and vote at the end.

These techniques fall outside of the scope of our research, which focuses on the *expressivity* of collaborative and multi-agent reasoning strategies. In contrast, these strategies fundamentally leverage the inherent *stochasticity* of trained LLMs to reduce noise and overcome failures. However, these techniques do not increase the intrinsic expressivity of the multi-agent system (e.g., using multiple agents to solve the same state tracking problem and performing voting may reduce error rate, but does not increase the worst-case expressivity of the model.). We further take this up in Appendix H.

(**vii**) *Some of the protocols described in the theory seem quite intricate. Can the protocols described in the theory be linked to protocols from the applied literature?*

Indeed, our results provide a rigorous foundation for certain approaches that have links to ideas from the applied literature. Related to the first regime in our theory (Section 4.2), in various studies, a single query is broadcast and worker agents each inspect only their local shard for an answer, returning a candidate and confidence to a lightweight aggregator that selects or fuses the outputs (e.g., (Zhou et al., 2025; Chang et al., 2025; Zhang et al., 2024a; Salve et al., 2024; Yang et al., 2025b). In our second regime (Section 4.3), recursive aggregation is optimal; this again is related to some existing approaches in the applied literature: agents iteratively compose partial states via tree-structured reduce/merge operators (e.g., parallel scan/prefix-sum and divide-and-conquer), so a global answer is assembled from local summaries with shallow, e.g. logarithmic, communication depth (e.g., (Kim & Kim, 2025; Zhou et al., 2025; Xiao et al., 2025; Xu et al., 2025)). In our third regime (Section 4.4), we prove sequential, multi-hop handoffs optimal; this again links to approaches where the system must iteratively query different agents' shards in a back-and-forth chain—passing intermediate facts as state—so communication depth scales with the hop count (e.g., Yang et al. (2025b); Liu et al. (2025); Nguyen et al. (2025); Wang et al. (2024a)).

Importantly, our results rigorously clarify in which regimes such communication protocols are optimal. Simultaneously, as discussed in Section 6, our results show how, in a rigorous sense, such more sophisticated protocols may improve over simpler but popular strategies such as chain-of-agents or majority voting.

# C PROOFS

## C.1 NOTATION

We denote with $\mathbb{N}$, $\mathbb{Z}$ and $\mathbb{R}$ the set of natural, integers and real numbers, respectively. We use bold letters for vectors (*e.g.* $\mathbf{v} \in \mathbb{R}^{d_1}$), bold uppercase letters for matrices (*e.g.* $\mathbf{M} \in \mathbb{R}^{d_1 \times d_2}$). All vectors considered are column vectors unless otherwise specified. The $i$-th row and the $j$-th column of a matrix $\mathbf{M}$ are denoted by $\mathbf{M}_{i,:}$ and $\mathbf{M}_{:,j}$. Let $\Sigma$ be a fixed finite alphabet of symbols, $\Sigma^*$ the set of all finite strings (words) with symbols in $\Sigma$ and $\Sigma^n$ the set of all finite strings of length $n$. We use $\varepsilon$ to denote the empty string. Given $p, s \in \Sigma^*$, we denote with $ps$ their concatenation.

## C.2 PROOFS FOR GENERAL RESULTS (SECTION 4.1)

**Proposition C.1** (Conservation of size, repeated from Proposition 4.1). *Any protocol can be converted into an equivalent single-agent protocol with the same size up to constant factor.*

*Proof.* By constructing a single agent that alternates between simulating each of the agents of the original protocol, computed by a Transformer $T_{\text{SingleAgent}}$.

Recall that all agents have the same Transformer parameters, $T_{\text{MultiAgent}}$.

We extend $\Xi$ with tokens $\langle\texttt{AgentID}\rangle$, one for each agent ID. We assume some fixed arbitrary ordering over all agent IDs.

The tokens of the reasoning chain record for each step which agent it is associated with. It alternatingly consists of $\langle\texttt{AgentID}\rangle$ and $\langle\texttt{CoT/communication token}\rangle$. Say, the even positions are the CoT/communication tokens (the nodes of the original chains); the odd positions are agent indices.

Now, the transformer $T_{\text{SingleAgent}}$ alternates between two modes, depending on the position.

At an $\langle\texttt{AgentID}\rangle$ position, the transformer knows from the input which agent to simulate. It then restrict attention to those tokens that are relevant to this agent, and otherwise performs the computations done by $T_{\text{MultiAgent}}$. The top layer then outputs whatever $T_{\text{MultiAgent}}$ would have output.

In the other mode, $T_{\text{SingleAgent}}$ needs to figure out which agent's turn it is next: It is the first agent in the ordering which is higher than the last agent but hasn't terminated yet. To achieve this, first retrieve the last agent's last previous turn, using a single attention head. Then find the first non-TERMINATE action succeeding that turn; this gives us the agent whose turn it is. The transformer then outpts this agent's ID as the next token. $\square$

**Proposition C.2** (Repeated from Proposition 4.2). *Consider a task with a multi-agent system whose communication budget is $\mathcal{O}(1)$ in $N$ across all $w \in [N]$. Then this task has a single-agent CoT with depth (and hence size) $\mathcal{O}(1)$.*

*Proof.* We take $w$ towards $N$, so that the chunk size becomes 1. Assume that the communication budget is $\mathcal{O}(1)$ in $N$ across all $w \in [N]$. Then we can find a protocol with communication budget $\mathcal{O}(1)$ where the depth at $w = N$ is $\mathcal{O}(1)$, as each agent only has a bounded number of tokens to process, which can be hard-coded into a UHAT transformer.

Now we use this to construct a protocol computing the function at depth $\mathcal{O}(1)$ at $w = 1$. To realize this, we increase the Transformer's number of layers to immediately compute the relevant agent's $\mathcal{O}(1)$ computation steps when processing the input in the context window; we code the agent identifiers into the positional embeddings. The only exception here is in the communication edges; for this, the transformer flags places where it would emit a communication edge. After reading the full context, the transformer performs $\mathcal{O}(1)$ CoT steps to simulate the $\mathcal{O}(1)$ communication steps and re-simulating the affected agents. $\square$

We also present here a technical lemma from the work of Vardi et al. (2021) we will use for the MLPs in some of our constructions

**Lemma C.1** ((Adapted from Lemma 22 of Vardi et al. (2021)). *Let $T$ be a threshold circuit with $d$ inputs, $q$ outputs, depth $m$ and size $s$. There is a neural network $N$ with $q$ outputs, depth $m + 1$ and size $2s + q$, such that for every input $\mathbf{x} \in \{0, 1\}^d$ we have $N(\mathbf{x}) = T(\mathbf{x})$. Moreover, for every input $\mathbf{x} \in \mathbb{R}^d$ the outputs of $N$ are in $[0, 1]$.*

## C.3 PROOF FOR ASSOCIATIVE RECALL (SECTION 4.2)

**Lemma C.2.** *Given an input consisting of $N$ key-value pairs $(\mathbf{x}_i, \mathbf{y}_i)$, and a queried key $\mathbf{x}_{query}$, there exists a two layer Transformer with $\mathcal{O}(\log N)$ width which returns the associated value.*

*Proof.* We consider that the Transformer agent receives the following input

$$\mathbf{X} = \begin{bmatrix} \mathbf{x}_1 & \mathbf{y}_1 & \cdots & \mathbf{x}_n & \mathbf{y}_n & \mathbf{x}_{\text{query}} \end{bmatrix}^\top \in \mathbb{R}^{2n+1 \times d}, \tag{3}$$

where $\mathbf{x}_i$ is a vector representing the key and $\mathbf{y}_i$ is a vector representing the value in a sequence of key-value pairs. We assume that every key value $\mathbf{x}_i$ has a unique associated value $\mathbf{y}_i$. We use the following embeddings for tokens:

$$\mathbf{x}_i = [\mathcal{T}(i) \quad \mathcal{T}(\mathbf{y}_\emptyset) \quad P_1(t) \quad P_2(t)] \tag{4}$$

$$\mathbf{y}_i = [\mathbf{0} \quad \mathcal{T}(\mathbf{y}_i) \quad P_1(t) \quad P_2(t)], \tag{5}$$

where both vectors are of size $\log n + \log(n+1) + 2$, . The first $\log n$ dimensions are Johnson-Lindenstrauss (J-L) vectors (Johnson et al., 1984) with the property that $\langle \mathcal{T}(i), \mathcal{T}(j) \rangle \leq 1/4$ for $i \neq j$ and $\langle \mathcal{T}(i), \mathcal{T}(j) \rangle \geq 3/4$ for $i = j$. These are used to embed the position $i$ and perform retrieval. The set of $\log(n+1)$ dimensions are used to embed the semantic information of each value vector. These also satisfy the same property as above. Each unique value is embedded in a different vector. Moreover we also define $\mathcal{T}(\mathbf{y}_\emptyset)$ to be the embedding vector corresponding to "no value found" Finally the positional encodings are defined as $P_1(t) = \cos\frac{\pi t}{N}$ and $P_2(t) = \sin\frac{\pi t}{N}$. Here, $t$ corresponds to the index of each token s.t. $t \in [2n+1]$. Not that this should not be confused with $i$, which indexes key-value *pairs*. The proof uses a 2-layer transformer model. The first layer copies the $\mathcal{T}(\mathbf{y}_i)$ vector from $\mathbf{y}_i$ over to the corresponding $\mathbf{x}_i$. This is done with two heads which we index (L) and (R) (for left and right respectively. We set the following:

$$\mathbf{W}_Q^{(L)} = \mathbf{W}_Q^{(R)} = \mathbf{W}_K^{(R)} = [\mathbf{0} \quad \mathbf{I}_2]^\top \tag{6}$$

$$\mathbf{W}_K^{(L)} = [\mathbf{0} \quad \boldsymbol{\rho}_\theta]^\top \tag{7}$$

where $\boldsymbol{\rho}_\theta$ is the rotation matrix given by

$$\boldsymbol{\rho}_\theta = \begin{bmatrix} \cos\theta & \sin\theta \\ -\sin\theta & \cos\theta \end{bmatrix}, \tag{8}$$

and $\theta$ is defined as

$$\theta = -\frac{\pi}{N}, \tag{9}$$

where $N$ is the length of the full sequence. The three first matrices directly select the positional embeddings from the input, and the last matrix selects the positional embeddings and shifts them by 1. Finally, we set

$$\mathbf{W}_V^{(L)} = \begin{bmatrix} \mathbf{I}_n & \mathbf{0} & \mathbf{0} \\ \mathbf{0} & \mathbf{0} & \mathbf{0} \\ \mathbf{0} & \mathbf{0} & \mathbf{0} \end{bmatrix} \tag{10}$$

$$\mathbf{W}_V^{(R)} = \begin{bmatrix} \mathbf{0} & \mathbf{0} & \mathbf{0} \\ \mathbf{0} & \mathbf{I}_{|\mathcal{D}|} & \mathbf{0} \\ \mathbf{0} & \mathbf{0} & \mathbf{0} \end{bmatrix} \tag{11}$$

The MLP for the first layer trivially computes an identity map. For the second layer, we use a construction similar to that of the retrieval heads used in the proof for Prop. 4.7. In essence we select the first $\log n$ dimensions of the vectors with both key and query matrices. This gives us an attention matrix which selects key-value vector which is equivalent to the query and puts it in the last component. Then, the MLP for the second layer extracts the $\mathcal{T}(\mathbf{y}_i)$ vector. The output matrix mapping to tokens is defined s.t. each row is a J-L vector $\mathcal{T}(\mathbf{y}_i), \forall i \in [n+1]$. Thus when computing the dot product with the extracted J-L vector, the majority of the probability mass is put on the correct output token. If the key vector $\mathbf{x}_i$ was not in the agent's chunk, the last token should still encode $\mathcal{T}(\mathbf{y}_\emptyset)$, which corresponds in the output matrix to a special "query not found" token. Using arg max decoding, this thus provides the correct behavior.

**Note on MLPs** Both computing the identity map and selecting a constant number of values in a vector are tasks that are trivially computable by a TC0 circuit. By appealing to Lemma C.1, we can thus obtain RELU-FFNs that can also perform such operations. $\qquad\square$

**Lemma C.3.** *Given a sequence of tokens $\{x_1, \ldots, x_n\}$, where all but one $x_i = x_\emptyset$, there exists a one-layer Transformer which can retrieve $x_i \neq x_\emptyset$*

*Proof.* Consider the Transformer agent receives the following input

$$\mathbf{X} = \begin{bmatrix} \mathbf{x}_1 & \ldots & \mathbf{x}_n \end{bmatrix}^\top \in \mathbb{R}^{n \times d}, \tag{12}$$

where for some $i^* \in [n]$, $\mathbf{x}_{i^*}$ is a one-hot encoding of an entity in $\mathcal{D}$ and $\mathbf{x}_i = \mathbf{x}_\emptyset$ otherwise. We use the following embeddings for tokens:

$$\mathbf{x}_i = \begin{bmatrix} \mathcal{T}(i) & \mathcal{T}(\mathbf{x}_i) \end{bmatrix}, \tag{13}$$

where both vectors are of size $\log n + \log(n+1)$. The first $\log n$ dimensions are J-L vectors similar to those used in the proof for Proposition 4.7, with the property that $\langle \mathcal{T}(i), \mathcal{T}(j) \rangle \leq 1/4$ for $i \neq j$ and $\langle \mathcal{T}(i), \mathcal{T}(j) \rangle \geq 3/4$ for $i = j$. The second $\log(n+1)$ dimensions are J-L vectors corresponding an encoding of each possible entity plus an encoding for the "not found" token $x_\emptyset$. The construction is essentially the same as the second layer of Lemma C.2. The key difference being that there are just "value" vectors and no keys, thus the first layer which copies the one-hot vector for the value vectors to the key vectors is not necessary. □

**Proposition C.3.** *Given an input consisting of $N$ pairs $(x_i, y_i)$, and a query $x$, consider the task of retrieving the (unique) $y$ such that $(x, y)$ appears in the input. Assume that the input is partitioned disjointly into parts provided to $k$ agents, which also have access to the query. Then they can solve the task with depth $\mathcal{O}(1)$ and communication $\mathcal{O}(1)$.*

*Proof.* The proof follows from Lemmas C.2 and C.3. The protocol is as follows:

Consider a setup with $w$ agents, and an input consisting of $N$ key-value pairs. Thus each agent receives a contiguous non-overlapping chunk of size $N/w$ as well as the same queried $x$. Each agent searches their subset of key-value pairs. The agent that finds the correct key communicates the associated value to a manager agent. All other agents return a special "null" token indicating they did not find the key. The manager then searches through the worker outputs and returns the value.

The worker agents can be simulated using Lemma C.2, and the manager agent extracting the correct answer from all returned agent tokens can be simulated using Lemma C.3. □

### C.4 PROOFS FOR STATE TRACKING RESULTS (SECTION 4.3)

**Proposition C.4** ((Repeated from Proposition 4.4). *Any multi-agent system computing* PARITY *requires size $\Omega(N)$.*

*Proof.* By proposition 4.1, we know that any protocol computing PARITY can be converted into an equivalent single agent protocol with some CoT length $L$. By Theorem 4.2 of Amiri et al. (2025), we have that any UHAT CoT for PARITY has length $\Omega(N)$.

Thus we must have $L \in \Omega(N)$. □

**Proposition C.5** (Repeated from Prop 4.6). *Given a finite monoid $M$ and any number of agents ($w : \mathbb{N} \to \mathbb{N}$ with $w(N) \in [N]$), there exists a $\mathcal{O}(\log w(N) + \frac{N}{w(N)})$ depth and $\mathcal{O}(N)$ size multi-agent system computing state tracking on $M$ with communication budget $w(N)$.*

*Proof.* Let an input $x$ of length $N$ be given, where each symbol is an element of $M$. We assume for simplicity (otherwise padding) that $N$ is a multiple of the number $w$ of agents. We build a DAG as follows.

The context given to agent $j$ is $x_{1,j}, \ldots, x_{N/w,j}, \mathrm{EOS}$ where EOS is the end of sequence (EOS) token. The context length of the sequence given to each agent is thus $N/w + 1$ (chunk size plus EOS token).

For each agent $j$, we create nodes $n_{1,j}, n_{2,j}, \ldots, n_{N/w,j}$, with CoT edges $n_{i,j} \to n_{i+1,j}$ with $\{t, x_{1,j} \ldots x_{i+1,j}\}$.

An agent can use a call SEND$\sigma$, where SEND is a special token to transmit information to other agents. Without loss of generality, we assume that this command transmits the symbol $\sigma$ to the next agent with ID $j + 1$. The final agent, which we call the receiver, only receives information and

does not transmit. The protocol computes a prefix sum algorithm with branching factor 2: at the beginning of runtime, all agents compute the composition of their $N/w$ elements. Then the agents with odd indices $j$ send their result to those with even indices, who compute the composition of their result with that of their odd index neighbor and so forth in a prefix sum fashion.

We show this is implementable in UHAT with 3 heads and a single layer, with width $\mathcal{O}(\log N)$. Essentially we use 2 heads to extract the value of the monoid elements and then store them in the EOS token and use the MLP to perform the rest of the processing.

**Embeddings** We will use quasi-orthogonal vectors to keep track of the positions of different elements in the sequence. Formally, let $\mathcal{T}(1), \ldots, \mathcal{T}(2N/w + 1)$ be $2N/w + 1$ vectors of dimension $k = \mathcal{O}(\log N)$ such that $\langle \mathcal{T}(i), \mathcal{T}(j) \rangle \leq 1/4$ for $i \neq j$ and $\langle \mathcal{T}(i), \mathcal{T}(j) \rangle \geq 3/4$ for $i = j$. Such vectors can be obtained using the Johnson-Lindenstrauss Lemma. We define $E(\sigma)$ to be the embedding vector of some symbol $\sigma \in \Xi$. Embeddings have the following structure

$$E(\sigma) = [\text{ohe}(\sigma) \quad \text{ohe}(\sigma) \quad \mathcal{T}(i) \quad \mathbf{0} \quad \mathbf{0} \quad \text{SEND}], \tag{14}$$

where $\text{ohe}(\sigma) \in \{0,1\}^{|\Xi|}$ is the one hot encoding (OHE) of $\sigma \in \Xi$, $\mathcal{T}(i)$s are quasi orthogonal vectors, the two last dimensions are also of dimension $k$ and where $[\text{send}] \in \{0,1\}$ are flags which are set to 0 by default. Equally, we define the embedding of the end of sequence token as

$$E(\text{EOS}) = [\mathbf{0} \quad \mathbf{0} \quad \mathbf{0} \quad \mathcal{T}(1) \quad \mathcal{T}(2) \quad \text{SEND}] \tag{15}$$

**Construction for composition of monoid elements** The construction for composition requires one layer and three heads. The key idea of the construction is to use two heads to extract the two elements to be composed at a given timestep, then concatenate them in the embedding of the $ token. The MLP can then perform the composition, which it returns in the embedding of the last token. The third head is only there to copy back the remaining embedding values. For the first head, we would have the following key, query and value matrices:

$$\mathbf{W}_Q = \begin{bmatrix} \mathbf{0} \\ \mathbf{0} \\ \mathbf{0} \\ \mathbf{I} \\ \mathbf{0} \end{bmatrix} \qquad \mathbf{W}_K = \begin{bmatrix} \mathbf{0} \\ \mathbf{0} \\ \mathbf{I} \\ \mathbf{0} \\ \mathbf{0} \end{bmatrix} \qquad \mathbf{W}_V = \begin{bmatrix} \mathbf{I} & \mathbf{0} & \mathbf{0} & \mathbf{0} & \mathbf{0} \\ \mathbf{0} & \mathbf{0} & \mathbf{0} & \mathbf{0} & \mathbf{0} \\ \mathbf{0} & \mathbf{0} & \mathbf{0} & \mathbf{I} & \mathbf{0} \\ \mathbf{0} & \mathbf{0} & \mathbf{0} & \mathbf{0} & \mathbf{0} \\ \mathbf{0} & \mathbf{0} & \mathbf{0} & \mathbf{0} & \mathbf{0} \end{bmatrix} \tag{16}$$

The output of the attention layer is thus all zeros except for the embedding at the EOS symbol which would be

$$E(\text{EOS}) = [\text{ohe}(\sigma) \quad \mathbf{0} \quad \mathbf{0} \quad \mathcal{T}(i) \quad \mathbf{0} \quad \text{SEND}], \tag{17}$$

The construction for the second head is very similar, with the main differences being the query matrix has the all 0s and identity at the *last* block and the value matrix is like that of the previous head with the two last columns swapped. This would give us a similar sequence of all 0 vectors, except for the embedding at the EOS symbol which would be

$$E(\text{EOS}) = [\mathbf{0} \quad \text{ohe}(\sigma) \quad \mathbf{0} \quad \mathbf{0} \quad \mathcal{T}(i) \quad \text{SEND}], \tag{18}$$

The third head trivially computes the identity matrix (but with 0s at the EOS position) by using both key and query matrices to extract the J-L vectors found at the "third" embedding block. We then use the $\mathbf{W}_O$ matrix to select the relevant parts of out of each head. Once this is done, we use the MLP to compute composition.

**MLP** The MLP computes a map as defined below. If $\text{SEND} = 0$:

$$[\text{ohe}(\sigma_1) \quad \text{ohe}(\sigma_2) \quad \mathbf{0} \quad \mathcal{T}(i_1) \quad \mathcal{T}(i_2) \quad \text{SEND}] \mapsto$$
$$[\text{ohe}(\sigma_1) \circ \text{ohe}(\sigma_2) \quad \text{ohe}(\sigma_1) \circ \text{ohe}(\sigma_2) \quad \mathbf{0} \quad \mathcal{T}(i_1 + c) \quad \mathcal{T}(i_2 + c) \quad \text{SEND}],$$

where $c$ is the token count between the first token and the EOS token, with $\text{ohe}(\sigma) \circ \text{ohe}(\sigma) \mapsto \text{ohe}(\sigma)$ and $\mathbf{0} \mapsto \mathbf{0}$ in the last two J-L positions.

If $\mathcal{T}(i_2 + c) = \mathcal{T}(2N/w)$, the model computes this slightly different map:

$$[\text{ohe}(\sigma_{2N/w-1}) \quad \text{ohe}(\sigma_{2N/w}) \quad \mathbf{0} \quad \mathcal{T}(2N/w - 1) \quad \mathcal{T}(2N/w) \quad \text{SEND}] \mapsto$$
$$[\text{ohe}(\text{SEND}) \quad \text{ohe}(\sigma_{2N/w-1}) \circ \text{ohe}(\sigma_{2N/w}) \quad \mathcal{T}(2N + 1) \quad \mathcal{T}(2N/w + 1) \quad \mathbf{0} \quad \text{SEND}],$$

Thus at the next step of decoding the final vector would stay the same.

If the SEND flag is equal to 1, the MLP simply swaps the values in the first $|\Xi|$ dimensions with those in the second $|\Xi|$ dimensions. Thus, once it is time to communicate the model outputs $\text{SEND}\sigma$.

Such a map can be defined by a threshold circuit. Composition of elements from a finite monoid can trivially be evaluated constant time by framing the problem as constant lookup. The shifting of indices is also in TC0 as this is reducible to counting/addition which is known to be in TC0 Nguyen & Cook (2006). Finally, TC0 is closed under boolean combination. Thus performing the conditional routing of which circuit to used based on the flag value is also in TC0. Finally, by Lemma C.1, we know that if such a circuit exists, it can be converted into an MLP which is linear in the circuit parameters.

**Output matrix**   Every row of the output matrix is a OHE of one of the symbols in $\Xi$. The output matrix is a combined transformation which first selects the top $|\Xi|$ dimensions and uses the OHE vector found there to put a 1 at the underlying position in the output vocabulary vector. Only the last token is used for prediction.

**Receiving and sending communication**   We assume all agents decode synchronously. When an agent receives a symbol, the protocol takes the agent's last symbol, and appends the received symbol as well as a EOS token. For simplicity, we assume that each agent is given a fresh new context at the beginning of a new round of communication. The construction could easily be extended by adding a layer which zeros out the embedding values of vectors from the previous query. To make sure all the agents only send symbols at the appropriate time, one can easily change the number of J-L vectors which the agent receives as these decide at what point the agent sends information.   □

**Proposition C.6.** *Let $L$ be a regular language over $\Sigma$. For each $w : \mathbb{N} \to \mathbb{N}$ ($w(N) \in [N]$), there is a multi-agent system with $w(N)$ agents that computes membership in $L$.*

*Proof.* This statement follows immediately as a consequence of Proposition 4.6.   □

**Proposition C.7** (Optimality (Repeated from Prop. 4.7))**.** *Assume the finite monoid $M$ is a nontrivial group, and $\mathcal{A}$ a multi-agent system computing state tracking over $M$. Then $\mathcal{O}(w(N))$ communication budget, and computation depth $\Omega(\frac{N}{w(N)})$ are each optimal.*

*Proof.* Optimality of the communication budget holds because each agent's portion matters for the result: each agent must send at least one message, hence the communication budget scales linearly with the number of agents. The computation depth lower bound follows from the Proposition 4.1 on size conservation:

$$N = Size \leq \text{Computation-Depth} \cdot \text{Agents} \tag{19}$$

hence

$$\frac{N}{w(N)} \leq \text{Computation-Depth.} \tag{20}$$

From which the result follows.   □

C.5   PROOFS FOR $k$-HOP REASONING (SECTION 4.4)

**Proposition C.8.** *Let the number of agents be $w : \mathbb{N} \to \mathbb{N}$ ($w(N) \in [N]$). The $k$-hop composition task with $N$ facts can be solved with computation depth $\mathcal{O}(k)$, communication budget $\mathcal{O}(k)$, and size $\mathcal{O}(w(N) \cdot k)$. The communication budget is optimal. The computation depth $\mathcal{O}(k)$ is optimal at least up to a $\log(N + k)$ factor.*

*Proof.* We start by exposing the communication protocol which we prove optimal. Then, we give the constructions of the worker and manager agents which implement this protocol. The protocol is as follows:

Let $N$ be the number of total key-value pairs in the context and let $w$ be the number of worker agents. Each worker agent receives a chunk $N/w$ as well as first query $f_1(\mathbf{x}_{\text{query}})$. In the first round, the each agent searches for $f_1(\mathbf{x}_{\text{query}})$ in its chunk. The agent that finds the queried key

communicates its associated value to a manager agent. The manager agent then updates the query s.t. $f_2(f_1(\mathbf{x}_{\text{query}})) = f_1(\mathbf{x}'_{\text{query}})$ and broadcasts this new queried key to all worker agents. This process repeats $k$ times in total until the entire $k$-hop query is resolved.

**Worker agent construction**   The worker agent essentially performs recall on a series of key-value vectors, letting

$$\mathbf{X} = [f(\mathbf{x}_1) \quad \mathbf{y}_1 \quad \cdots \quad f(\mathbf{x}_n) \quad \mathbf{y}_n \quad f(\mathbf{x}_{\text{query}})]^\top \in \mathbb{R}^{2n+1 \times d}, \tag{21}$$

we can straightforwardly apply Lemma C.2. We note that the same agent can be used across hops; if we simply append the new queried key to the end current sequence, the worker construction will return the value for the rightmost queried key. This is true because the construction uses rightmost tie-breaking in attention.

**Manager construction**   The manager agent receives a sequence of functions

$$[\mathbf{f}_1 \quad \cdots \quad \mathbf{f}_n \quad \mathbf{y}_{\text{query}} \quad E(\#)], \tag{22}$$

where $\mathbf{f}_i$ is an embedding of the function that sends an entity to another through their relationship, $\mathbf{y}_{\text{query}}$ is the current known entity and $E(\#)$ is the embedding of the EOS token. The key idea of the manager construction is to compose together the last function with the query entity in order to return a new entity value. To do so, we define a transition map as

$$\mathbf{q}_i = \sum_{x \in \mathcal{D}} f(x)\mathbf{e}_{i_x} \in \mathbb{R}^{|\mathcal{D}|}, \tag{23}$$

with the full embedding vector being

$$\mathbf{f}_i = [\mathbf{q}_i \quad \mathbf{0} \quad \mathcal{T}(i) \quad \mathbf{0}] \in \mathbb{R}^{2|\mathcal{D}|+2\log N} \tag{24}$$

where $\mathbf{e}_{i_x}$ is the canonical basis vector corresponding to the entity $x$ for some indexing $\mathcal{I}$. Consequently, we have that $\mathbf{y}_{\text{query}} = \begin{bmatrix} \mathbf{0} & \mathbf{e}_{i_y} & \mathcal{T}(n+1) & \mathbf{0} \end{bmatrix} \in \mathbb{R}^{2|\mathcal{D}|+2\log N}$ i.e. a canonical basis vector with a one in the position of the corresponding entity in the second half of the vector. Similarly to the worker agent construction, we define the embedding of EOS token as

$$E(\#) = [\mathbf{0} \quad \mathbf{0} \quad \mathcal{T}(n) \quad \mathcal{T}(n+1)] \tag{25}$$

where $\mathcal{T}(1), \ldots, \mathcal{T}(n+1)$ are J-L vectors s.t. $\langle \mathcal{T}(i), \mathcal{T}(j) \rangle \leq 1/4$ for $i \neq j$ and $\langle \mathcal{T}(i), \mathcal{T}(j) \rangle \geq 3/4$ for $i = j$. The attention layer is defined in a similar manner to that of the one in the proof for Proposition 4.7 and uses two heads to retrieve both the $n$th and $n + 1$th elements in the sequence using the positional encoding given through J-L vectors.

The MLP then computes the composition of $\mathbf{q}_n$ with $\mathbf{e}_{i_y}$ and outputs the OHE vector of the resulting token. This can easily be done leveraging Lemma 5 of Liu et al. (2022).

**Optimality**   The protocol described above has size $\Theta(wk)$, communication budget $\Theta(k)$, and computation depth $\Theta(k)$.

Optimality of the communication budget follows because composition of $k$ permutations over $\{1, \ldots, 5\}$ has communication complexity $\Omega(k)$ in the model where one agent has the even positions and the other the odd positions (Tesson & Thérien, 2002). Thus, communication budget is $\Omega(k)$ even when $w(N) \equiv 2$. Extension to larger $w(N)$ follows by considering the case where all relevant facts happen to be distributed between two agents.

To prove that the depth is worst-case optimal up to a logarithmic factor, we consider the case where all relevant facts happen to be distributed between two agents. Hence, these two agents must jointly emit $\Omega(k)$ communication bits. Because an agent emits only $\mathcal{O}(\log |\Xi_{N+k}|) = \mathcal{O}(\log(N + k))$ bits at a step of time, the number of communication steps between these two agents (and hence the computation depth) must be lower-bounded by $\Omega(\frac{k}{\log(N+k)})$. $\qquad\qquad\square$

# D  PARITY ERROR ANALYSIS

**Definitions.** Fix integers $k \geq 2$ and $c \geq 1$, and set $N := k^c$. The Parity function $P : \{0,1\}^m \to \{0,1\}$ is defined as

$$P(x_1, \ldots, x_m) := x_1 \oplus \cdots \oplus x_m,$$

where $\oplus$ denotes XOR or addition in $\mathbb{F}_2$. Let $f : \{0,1\}^k \to \{0,1\}$ be a randomized algorithm which takes input $x$ of length exactly $k$ and outputs $P(x)$ with probability $1 - \epsilon$, and outputs $1 - P(x)$ with probability $\epsilon$. In other words, it computes the Parity function with error $\epsilon$. The function $f(x)$ can also be written as,

$$f(x) = P(x) \oplus e,$$

where the *node error* $e \in \{0,1\}$ equals 1 with probability $\varepsilon$ and 0 with probability $1 - \varepsilon$. Assume different calls of $f$ are independent (hence their error bits are i.i.d. Bernoulli($\varepsilon$)).

Given an input $z \in \{0,1\}^N$, define the composite algorithm $F$ where we first divide the input $z$ in $N/k$ chunks of length $k$, and apply $f$ on each of them to get $N/k$ outputs. We do this recursively to obtain the final output. The computation can be seen as a $k$-ary tree of depth $c$ with $N$ leaves (the bits of $z$); at each internal node, apply $f$ to the $k$ outputs of its children. The output $F(z)$ is the bit at the root. The tree has $M$ nodes where

$$M = \sum_{j=0}^{c-1} k^j = \frac{N-1}{k-1}.$$

Each internal node is computed by an independent call to $f$.

**Lemma D.1** (Error of a parity tree with i.i.d. flips)**.** *For every fixed $z \in \{0,1\}^N$, let error of $f$ be $\epsilon$, the probability that the algorithm $F$ makes an error is given by,*

$$\Pr\left[F(z) \neq P(z)\right] = \frac{1 - (1 - 2\varepsilon)^M}{2}, \qquad M = \frac{N-1}{k-1}.$$

Thus the probability of making a mistake using the Prefix sum protocol on a string of length $N$, assuming a branching factor of $k = 2$ would be

$$\Pr\left[F(z) \neq P(z)\right] = \frac{1 - (1 - 2\varepsilon)^{N-1}}{2}$$

Note that as $N \to \infty$, we asymptotically reach a probability of 1/2, even for very small $\varepsilon$. However, for small $k$ values, the number of tokens processed by each each is also very small and thus the performance only degrades for very large $N$. This is the regime in which we operate in practice, thus corroborating this theoretical model to our experimental results.

*Proof.* We first show that the algorithm $F$ makes an error if and only if there are odd number of errors in the nodes of the computation tree.

For each node $u$, let $s(u)$ denote the true parity of the leaves in the subtree of $u$, and let $\widehat{s}(u)$ be the value computed by $F$ at $u$. Define the node error indicator $\delta(u) := \widehat{s}(u) \oplus s(u) \in \{0,1\}$. For an internal node $v$ with children $u_1, \ldots, u_k$, write $e_v$ for the (independent) error bit of the call to $f$ at $v$, so that

$$\widehat{s}(v) = f\left(\widehat{s}(u_1), \ldots, \widehat{s}(u_k)\right) = P\left(\widehat{s}(u_1), \ldots, \widehat{s}(u_k)\right) \oplus e_v.$$

Using linearity of $\oplus$,

$$\delta(v) = \widehat{s}(v) \oplus s(v) = \left(\bigoplus_{i=1}^{k} \widehat{s}(u_i) \oplus e_v\right) \oplus \left(\bigoplus_{i=1}^{k} s(u_i)\right) = \left(\bigoplus_{i=1}^{k} \delta(u_i)\right) \oplus e_v.$$

Since $\delta(\text{leaf}) = 0$, induction up the tree yields $\delta(\text{root}) = \bigoplus_v e_v$. Thus, the algorithm $F$ errs if and only if an odd number of node calls make an error.

Let $S = \delta(\text{root}) = \bigoplus_v e_v \in \{0,1\}$ be the random variable which takes the value 1 when $F$ makes an error and is 0 otherwise. We introduce another random variable $Y_v = (-1)^{e_v} \in \{-1, +1\}$. And let $Y := (-1)^S = \prod_v Y_v$ by rewriting Parity as a product, which is quite standard to make calculations convenint.

Since the calls are independent, we have

$$\mathbb{E}[Y] = \prod_v \mathbb{E}[Y_v] = \big((1-\varepsilon)\cdot 1 + \varepsilon\cdot(-1)\big)^M = (1-2\varepsilon)^M.$$

Since $Y = +1$ iff $S = 0$ and $Y = -1$ iff $S = 1$,

$$\mathbb{E}[Y] = \Pr[S = 0] - \Pr[S = 1], \qquad \Pr[S = 0] + \Pr[S = 1] = 1.$$

Solving the two equations above gives

$$\Pr[S = 1] = \Pr[F(z) \neq P(z)] = \frac{1 - (1-2\varepsilon)^M}{2}.$$

$\square$

# E EXPERIMENTAL DETAILS

General details:

- All experiments were run on TogetherAI API
- Models used: Llama-3.1-8B-Instruct-Turbo, Llama-3.3-70B-Instruct-Turbo, EXAONE-3.5-32B-Instruct
- All experiments are run 100 times with a seed set to 42 for consistency.
- Three multi-agent architectures tested: Majority Voting, Chain-of-Agents, and Prefix Sum
- For all experiments, examples are generated on the fly given the length and difficulty parameters specified in the script.

Throughout, we use 8 agents in the majority voting setup. We ablated over [2, 4, 8, 16] agents across all considered tasks and found that past 8 agents, there was no significant improvement in any of the tasks. The choices for task-specific hyperparameters are discussed in their respective subsections.

## E.1 ASSOCIATIVE RECALL TASK

Key-value strings are generated at random. A recall query is sampled uniformly from the keys. Models are prompted their roles (manager or worker) explaining what they must do and the communication is handled deterministically.

In experiments, we test sequence lengths (i.e., the number of key-value pairs) as powers of two ranging from $2^4$ to $2^{11}$. We use 8 agents for Majority Voting. For Chain-of-Agents, we select the optimal chunk size—chosen from powers of two between 8 and 64—for each sequence length.

---

**Associative Recall Majority Vote Agent Prompt**

You are a reasoning agent responsible for analyzing a portion of a document. Your task is to detect a specific value in a sequence of key-value pairs, given a corresponding key. Follow these steps:

1. Identify if the key is present in the sequence of key-value pairs.

2. If the key is present, return the value corresponding to the key.

3. If the key is not present, return "NOT_FOUND".

4. Present the final answer in the format "The answer is: [your answer]"

You MUST use the following template. ONLY OUTPUT THE ANSWER. Here is an example for "23 42 12 34 56 78 90 12 | Query: 56":

```
The answer is: 78
```

---

---

**Associative Recall Chain-of-Agents Worker Prompt**

You are a reasoning agent responsible for analyzing a portion of a document. Your task is to detect a specific value in a sequence of key-value pairs, given a corresponding key. Follow these steps:

1. Identify if the key is present in the sequence of key-value pairs.

2. If the key is present, return the value corresponding to the key.

3. If the key is not present, return "NOT_FOUND".

4. Present the final answer in the format "The answer is: [your answer]"

You MUST use the following template. ONLY OUTPUT THE ANSWER. Here is an example for "23 42 12 34 56 78 90 12 | Query: 56":

```
The answer is: 78
```

---

**Associative Recall Chain-of-Agents Manager Prompt**

You are a manager agent responsible for synthesizing information from multiple workers. Your task is to combine their provided values and determine the value corresponding to the query. To compute the final value, follow these steps:

1. Collect the value results from all worker agents.

2. Each worker will return either a value or "NOT_FOUND".

3. Exactly one worker will return the value corresponding to the query, the rest will return "NOT_FOUND".

4. Report the value corresponding to the query as your output.

5. Present the final answer in the format "The answer is: [your answer]"

You MUST use the following template. ONLY OUTPUT THE ANSWER. Here is an example for "NOT_FOUND NOT_FOUND 78 NOT_FOUND":

```
The answer is: 78
```

---

## E.2 Parity Calculation Task

String of bits of fixed length are sampled uniformly at random. Ground truth is computed with a function which evaluates parity. Models are prompted their roles (manager or worker) explaining what they must do and the communication is handled deterministically.

The experiments test sequence lengths as powers of 2, ranging from $2^4$ to $2^9$ (i.e., length 8) with support for index hints to aid model reasoning. We use Majority Voting over a total of 8 agents. For Chain-of-Agents, binary strings are split into chunks of size 8, while Prefix Sum uses a branching factor of 4 for hierarchical processing. For Chain-of-Agents, the optimal chunk size was determined by ablating over the range [2, 4, 8, 16]. For Prefix Sum, the optimal branching factor was determined by ablating over the range [2, 4, 8]. The task evaluates models' ability to accurately count 1-bits and determine even/odd parity across different architectural approaches.

The data for the Pareto frontier plots (computation depth vs. total communication) was generated by ablating over the branching factor for the prefix sum protocol. The total communication (number of edges) is can be computed straightforwardly as the sum of all edges of the log-depth tree with the corresponding branching factor. Sequence lengths are taken to be powers of two. For each sequence length $2^n$, we ablate over powers of two from 2 to $2^{n-1}$.

---

**Parity Majority Vote Agent Prompt**

You are a reasoning agent responsible for analyzing a portion of a document. Your task is to provide an analysis of the binary string provided in your chunk and determine if it is even or odd parity. To compute the parity, follow these steps:

1. Count the number of 1's in the binary string.

2. If the count is even, return 0.

3. If the count is odd, return 1.

---

4. Present the final answer in the format "The answer is: [your answer]"

You MUST use the following template. Here is an example for "1011":

```
1: 1 (count: 1)
2: 0 (count: 1)
3: 1 (count: 2)
4: 1 (count: 3)
Final count: 3
The answer is: 1
```

## Parity Prefix Sum Prompt

You are a manager agent responsible for synthesizing the results of previous workers. Your task is to return the parity of the binary string provided by the worker agents. You may think step by step, but your final answer should be concise and clear. To compute the parity, follow these steps:

1. Collect the results from the worker agents. This should be a list of binary digits (0 or 1).
2. If the parity of the list is even, return 0.
3. If the parity of the list is odd, return 1.
4. Present the final answer on a new line in the format "The answer is: [your answer]"

IMPORTANT: Show your work step by step to demonstrate thorough analysis:

1. Go through each bit position and note its value
2. Keep a running count of 1s encountered
3. State the final count
4. Determine if the count is even or odd

You MUST use the following template. Here is an example for "1011":

```
1: 1
0: 1
1: 2
1: 3
Final count: 3
The answer is: 1
```

## Parity Chain-of-Agents Worker Prompt

You are a worker agent responsible for analyzing a portion of a document. Your task is to provide an analysis of the binary string provided in your chunk and determine if it is even or odd parity. To compute the parity, follow these steps:

1. Count the number of 1's in the binary string.
2. If the count is even, return 0.
3. If the count is odd, return 1.
4. Provide your result in a clear and concise manner.
5. Present the final answer in the format "The answer is: [your answer]"

You MUST use the following template. Here is an example for "1011":

```
1: 1
0: 1
1: 2
1: 3
Final count: 3
The answer is: 1
```

---

**Parity Chain-of-Agents Manager Prompt**

You are a manager agent responsible for synthesizing information from multiple workers. Your task is to combine their provided parities and determine the overall parity of the binary string. To compute the aggregate parity, follow these steps:

1. Collect the parity results from all worker agents.

2. Each worker will return either 0 or 1.

3. Count the number of 1 responses.

4. If the count of 1 responses is even, the overall parity is 0.

5. If the count of 1 responses is odd, the overall parity is 1.

6. Present the final answer in the format "The answer is: [your answer]"

You MUST use the following template. Here is an example for "1011":

```
1: 1
0: 1
1: 2
1: 3
Final count: 3
The answer is: 1
```

---

### E.3 $S_5$ PERMUTATION TRACKING TASK

We frame the $S_5$ permutations task as a word problem where each agent is given a prompt explaining there are 5 balls in 5 distinct bins and a sequence of swap commands such as "swap ball 1 and 3, swap ball 2 and 4". In this task the agents must return the correct value of the ball in each bin. The bin numbers are only given at the beginning of the task making this a *hard* state tracking problem (Merrill et al., 2024).

The experiments test permutations with varying numbers of swaps ranging from 4 to 12 swaps with a step size of 2. By default, the task is constrained to $A_5$ (even permutations only) by forcing an even number of swaps. For Chain-of-Agents, swap sequences are processed in chunks of 2 swaps per worker, while Prefix Sum uses a branching factor of 2 with each worker handling exactly one swap operation. For both Chain-of-Agents and Prefix Sum, we tuned on the range [2,4] for the chunk size/branching factor respectively. Given the small sequence lengths used in this task, larger chunk size/branching factor options were not feasible. The task evaluates models' ability to maintain accurate state tracking through sequential ball position updates.

---

**Permutation Majority Vote Agent Prompt**

You are a reasoning agent responsible for tracking ball positions through a sequence of swaps.
Your task is to determine the final position of each ball after performing all the given swap operations.
Initial state: Each ball starts in its corresponding bin (ball 1 in bin 1, ball 2 in bin 2, etc.).
To solve this problem:

1. First, identify which balls are mentioned in the swap operations - ONLY track these balls

2. Start with balls in their initial positions (e.g., if balls 1, 2, 3 are mentioned: {1:1, 2:2, 3:3})

3. For each swap operation "Swap ball X and ball Y":
   - Find the current bins of ball X and ball Y
   - Exchange their positions

4. Continue until all swaps are processed

5. Present your final answer as a dictionary mapping ONLY the balls mentioned in swaps to their final positions

IMPORTANT: Only include balls that appear in the swap operations. Do not add extra balls.
Present the final answer in the format "The answer is: {ball1:bin1, ball2:bin2, ...}" for only the balls involved in swaps.

---

### Permutation Chain-of-Agents Worker Prompt

You are a worker agent responsible for processing a portion of swap operations in a larger sequence. Your task is to carefully track ball positions through your assigned swap operations and report the precise current state.
You will receive:

- Current ball positions as a dictionary (e.g., {1:2, 2:1, 3:3})
- A sequence of swap operations to process

Instructions:

1. Start with the EXACT positions given to you - this is the state after previous swaps
2. Process each swap operation "Swap ball X and ball Y" in order:
    - Find the current bins of ball X and ball Y
    - Exchange ONLY their positions
    - Keep all other balls in their current positions
3. Track each swap carefully - one mistake will affect the final result
4. Report the state after processing ALL your assigned swaps

CRITICAL: Only include the balls that are present in the input positions. The exact same balls, no more, no less.
Present the final answer in the format "The answer is: {ball1:bin1, ball2:bin2, ...}" with the exact same ball numbers as your input.

### Permutation Chain-of-Agents Manager Prompt

You are a manager agent responsible for determining the final ball positions from worker results.
Your task is to identify the final state of all balls after all swap operations have been processed by your workers.
You will receive position dictionaries from multiple workers who processed different portions of the swap sequence in order. The workers processed swaps sequentially, so:
Instructions:

1. The workers processed swaps in chronological order (worker 1 → worker 2 → worker 3, etc.)
2. Each worker started with the positions left by the previous worker
3. The LAST worker's result contains the final positions after all swaps
4. Simply report the last worker's position dictionary as the final answer

CRITICAL: Take the position dictionary from the last (final) worker only. This represents the complete final state.
Present the final answer in the format "The answer is: {ball1:bin1, ball2:bin2, ...}" exactly as reported by the final worker.

### Permutation Prefix Sum Worker Prompt

You are a worker agent in a hierarchical system processing ONE swap operation.
IMPORTANT: Initially, balls start in their corresponding bins (ball 1 in bin 1, ball 2 in bin 2, etc.). Through swaps, balls can move to different bins.
Your task is to apply exactly one swap operation and report the resulting ball positions with perfect accuracy.
You will receive:

- Current ball positions as a dictionary (e.g., {1:3, 2:1, 3:2, 4:5, 5:4})
- ONE swap operation: "Swap ball X and ball Y"

Process the swap step-by-step using this exact reasoning template:

1. Current state: [copy the input dictionary]
2. Operation: [copy the swap operation]
3. Ball X is currently in bin: [identify bin number]

4. Ball Y is currently in bin: [identify bin number]

5. After swap: Ball X moves to bin [Y's old bin], Ball Y moves to bin [X's old bin]

6. Verification: Check that only these two balls changed positions, all others remain the same

7. Final state: [complete updated dictionary]

CRITICAL CONCEPT: You are tracking which BALL is in which BIN.

- BALLS are the moving objects (numbered 1, 2, 3, 4, 5)

- BINS are the fixed locations (numbered 1, 2, 3, 4, 5)

- When you swap "ball X and ball Y", you move those balls to different bins

- The bins stay in place - only the balls move between them

CRITICAL: Include ALL balls from input with exact same ball numbers. One swap affects exactly two positions.

Present the final answer in the format "The answer is: {ball1:bin1, ball2:bin2, ...}" with all balls from your input.

---

### Permutation Prefix Sum Manager Prompt

You are a manager agent in a hierarchical ball-tracking system combining results from workers.
IMPORTANT: Initially, balls start in their corresponding bins (ball 1 in bin 1, ball 2 in bin 2, etc.). Through swaps, balls move to different bins.
Your task is to determine the final ball positions after your workers processed their assigned swaps in sequence.
You will receive position dictionaries from workers who processed swaps in chronological order. Each worker:

- Started with the ball positions left by the previous worker

- Applied exactly one swap operation

- Reported the updated positions

Your job requires careful validation and explicit reasoning:

1. Validate that each worker's result is a logical continuation of the previous worker's output

2. Show the complete sequence of states from initial to final

3. Verify that each step represents exactly one swap operation

4. Report the last worker's result as your final output

Use this reasoning template:

1. Initial state: [first worker's input state]

2. After worker 1: [worker 1's result] - validate this is one swap from initial

3. After worker 2: [worker 2's result] - validate this is one swap from worker 1's result

4. Final result: [last worker's result]

CRITICAL: Output exactly the position dictionary from the final (last) worker. This contains the cumulative effect of all swaps.
Present the final answer in the format "The answer is: {ball1:bin1, ball2:bin2, ...}" exactly as reported by the last worker.

## E.4 $k$-HOP REASONING TASK

We create a list of entities and a list of relations. We create the fact base by sampling at random two entities and a relation and then deterministically generating a string. This procedure is done as many times as the number of facts needed. Then relationships are sampled in order to form a valid query that has its answer in the fact base.

The experiments test the models' ability to follow multi-step reasoning chains with varying numbers of hops (relationships to traverse). We use 50 balanced single-token entity names (25 male, 25 female) and 20 diverse single-token relations (boss, instructor, teacher, etc.). For this task, we generate problems with $k$ hops where $k$ ranges from 4 to 20 hops with a step size of 2. The experiment was

repeated with 100, 200 and 500 total fact count. For the IterativeQueryAgents approach, facts are divided into chunks of 20 facts per worker. Across all three regimes, we ablated over the range [10, 20, 50] and found 20 to yield the best accuracy.

---

### K-hop Majority Vote Agent Prompt

You are an expert at logical reasoning and following relationship chains.
Your task is to answer questions about relationships between people by following chains of connections through the given facts.
You will be given:

1. A set of facts describing relationships between people (e.g., "Alice's boss is Bob")

2. A query asking about a multi-step relationship chain

Instructions:

- Read all the facts carefully

- Follow the relationship chain step by step

- Track each connection to find the final answer

- Output your answer in the exact format: Answer: [PersonName]

Example: Facts: "John's boss is Mary. Mary's supervisor is Tom." Query: "Who is the supervisor of the boss of John?" Reasoning: John's boss is Mary → Mary's supervisor is Tom Answer: Tom
Be systematic and double-check your reasoning chain.

---

### K-hop IterativeQuery Worker Agent Prompt

You are a helpful assistant that answers questions based ONLY on the given facts.
IMPORTANT: You have been given only a small subset of all available facts. It is very likely that the fact needed to answer the query is NOT in your subset.
You will be given:

1. A limited set of facts about relationships between people

2. A specific query about one relationship

Instructions:

- ONLY look through the facts provided to you

- If you find the EXACT fact needed to answer the query, extract the answer

- If the exact fact is NOT in your subset (which is very common), respond with "Not Found"

- DO NOT guess or infer answers from similar facts

- DO NOT make assumptions about relationships not explicitly stated

- DOUBLE-CHECK: Before giving your final answer, carefully re-read the facts to ensure you have the correct match

- Always format your response as: Answer: [YourAnswer]

Example (found): Facts: "John's boss is Mary. Alice's teacher is Bob." Query: "Who is John's boss?" Response: Answer: Mary
Example (not found - very common): Facts: "John's boss is Mary. Alice's teacher is Bob." Query: "Who is Sarah's mentor?" Response: Answer: Not Found
Example (not found - don't guess): Facts: "John's boss is Mary. Alice's teacher is Bob." Query: "Who is Mary's supervisor?" Response: Answer: Not Found
CRITICAL: Before responding, double-check your work:

1. Re-read the query to understand exactly what is being asked

2. Scan through ALL the facts again to verify your answer or confirm it's not found

3. Make sure the relationship type matches exactly (e.g., "boss" vs "supervisor")

4. Only provide an answer if you are completely certain it appears in the facts

Remember: Most queries will not have their answer in your subset of facts. Only answer if the exact fact is present and you have double-checked it.

---

**K-hop IterativeQuery Manager Agent Prompt**

You are a manager agent that coordinates multi-hop reasoning queries.
Your task is to take an answer from a previous query and generate the next query in the reasoning chain.
You will be given:

1. The original multi-hop question

2. The current intermediate answer

3. The current step number

Instructions:

- Use the intermediate answer to construct the next query

- Format your response as: Next Query: [YourQuery]

Example: Original question: "Who is the supervisor of the boss of John?" Current answer: "Mary" (John's boss) Response: Next Query: Who is Mary's supervisor?

---

### E.5    NEEDLE-IN-A-HAYSTACK

For the needle-in-a-haystack test, we follow the implementation given by Kamradt (2024). We use as query: "What is the best thing to do in San Francisco?", with associated answer "The best thing to do in San Francisco is eat a sandwich and sit in Dolores Park on a sunny day."). The text in which the needle is embedded is the corpus of Paul Graham essays ( 14k words). For context lengths longer than 14k words, we simply concatenate the corpus multiple times until we achieve the desired length. The original implementation does the same. We repeat each experiment 10 times and report average accuracy across 10 runs. For Majority Voting, we use 8 agents, for CoA, we use a chunk size of 2000. Agent count is chosen from the range [3,5,8] and chunk size was chosen between [200, 1000, 2000]. Given the prohibitive number of datapoints for the full heatmap, we tuned hyperparameters on a subset containing only the 4 last context lengths and depths from 0% to 50% as those seemed, in the literature, to be the most problematic datapoints. We use Meta-Llama-3.1-8B-Instruct-Turbo for all considered experiments.

## F    ADDITIONAL EXPERIMENTS

In this section, we provide experiments from tasks and models that are not featured in the main paper.

### F.1    RECALL

Figure 6 shows accuracy for different sequence lengths. For shorter sequences (64–512), performance is similar, with Majority Voting sometimes outperforming CoA, but the multi-agent approach gains an edge as length increases. This trend is consistent with theoretical understanding. Recall is a task easily solved by Transformers, even with limited CoT (Arora et al., 2023; Bhattamishra et al., 2024). Thus, at shorter sequence lengths, the communication overhead may be detrimental by e.g., leading to hallucinations in models that do not have the key-value pair in their context.

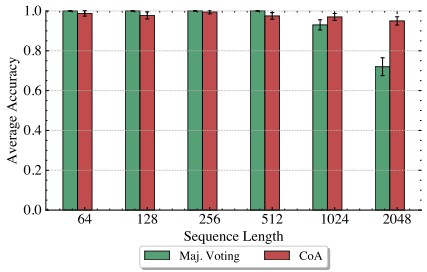

Figure 6: Llama-70B accuracy on RECALL across sequence lengths. CoA is the theoretically optimal protocol.

## F.2 PARITY

In this section, we provide PARITY results similar to those in the main text, but with llama-8B as the base model for agents. The plots here show trends similar to those in the main text. We note that llama-8B has poorer accuracy on the task; this is due to the deviating from the prompted guidelines and making mistakes (i.e. flipping bits) more frequently

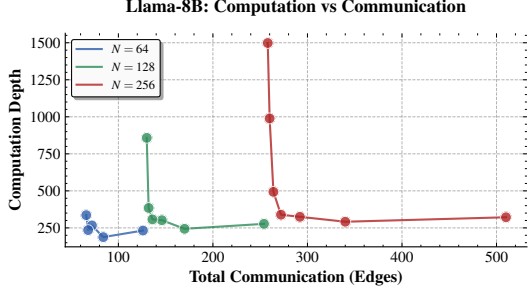

Figure 7: Communication vs Computation tradeoff for Llama-8B showing the relationship between communication budget and computation depth across different multi-agent protocols for the parity task.

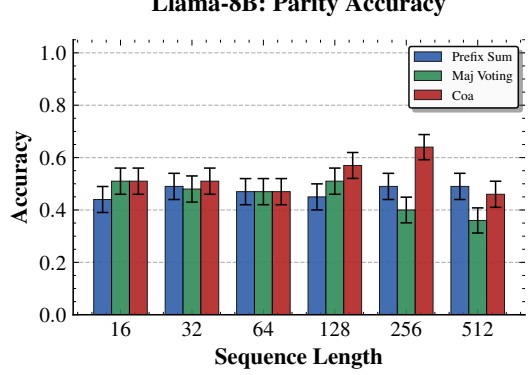

Figure 8: Parity calculation accuracy for Llama-8B across different sequence lengths, comparing single-agent vs multi-agent performance.

## F.3 $S_5$ PERMUTATIONS

Figures 9 and 10 provide detailed comparisons between Llama-8B and Llama-70B models across all three multi-agent approaches.

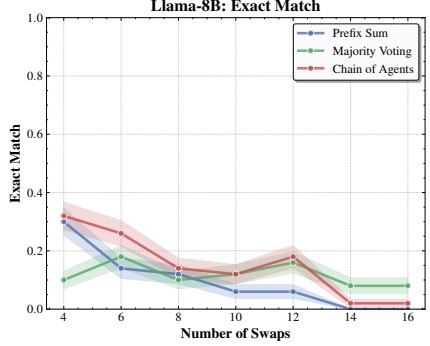

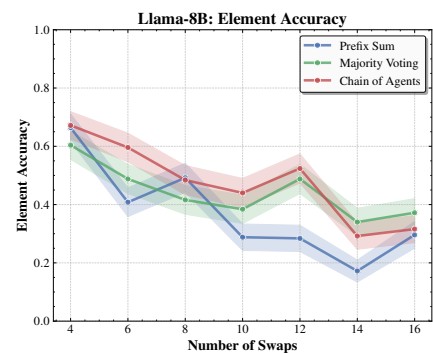

(a) Exact match accuracy for Llama-8B on the $S_5$ permutations task. Prefix Sum represents the theoretically optimal communication protocol, Majority Voting is self-consistency with majority voting decision, and CoA is Chain-of-agents protocol. Performance degrades as the number of swaps increases, with Prefix Sum maintaining superior performance.

(b) Per-element accuracy for Llama-8B on the $S_5$ permutations task. Element accuracy measures the fraction of correctly placed elements in the permutation. Shaded regions represent standard error bounds across multiple runs.

Figure 9: Performance comparison of multi-agent approaches on the $S_5$ permutations task using Llama-8B.

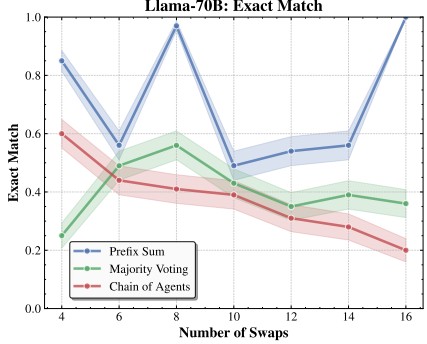

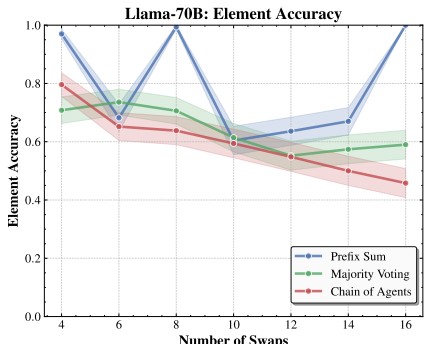

(a) Exact match accuracy for Llama-70B on the $S_5$ permutations task. The larger model demonstrates consistently improved performance across all agent types compared to Llama-8B, with the performance gap being most pronounced for the Chain of Agents approach.

(b) Per-element accuracy for Llama-70B on the $S_5$ permutations task. The improved reasoning capabilities of the larger model help mitigate the composition complexity challenges inherent to multi-agent coordination.

Figure 10: Performance comparison of multi-agent approaches on the $S_5$ permutations task using Llama-70B.

## F.4 $k$-HOP REASONING

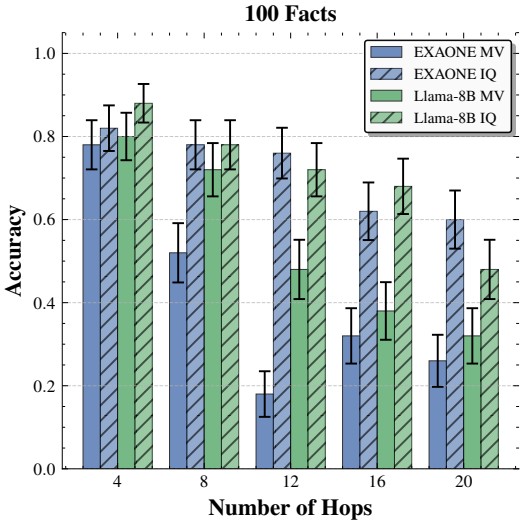

Figure 11: Comparison of multi-agent approaches for $k$-hop reasoning with 100 facts. Shows performance across different hop lengths for Llama-8B and EXAONE models.

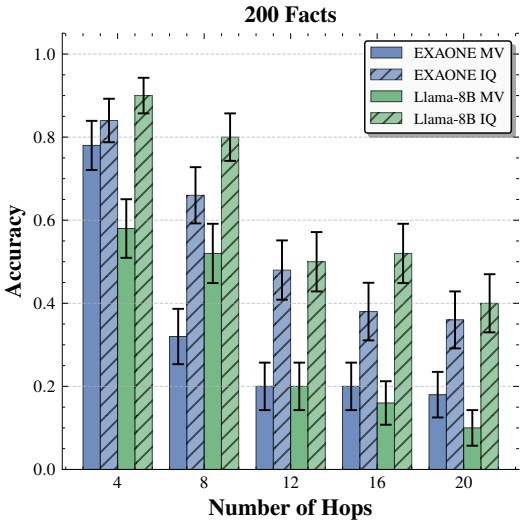

Figure 12: Comparison of multi-agent approaches for $k$-hop reasoning with 200 facts. Shows performance across different hop lengths for Llama-8B and EXAONE models.

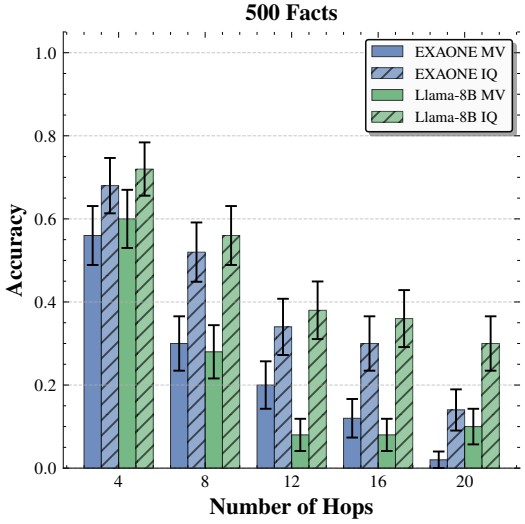

Figure 13: Comparison of multi-agent approaches for $k$-hop reasoning with 500 facts. Shows performance across different hop lengths for Llama-8B and EXAONE models.

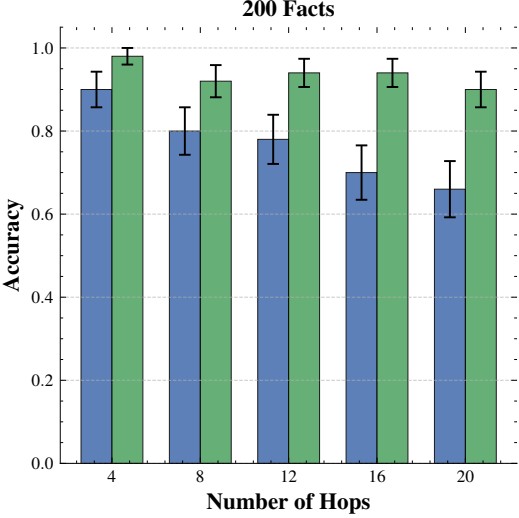

Figure 14: Llama-70B $k$-hop reasoning accuracy with 200 facts.

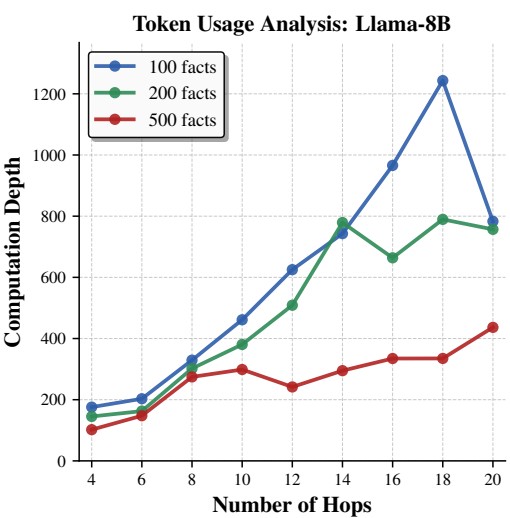

Figure 15: Computation depth vs. number of hops in the query for Lama-8B.

## G   BEYOND SINGLE-TOKEN COMMUNICATION

Definition 3.1 assumes that agents can only send and receive a single token at each time step. This assumption is for mathematical convenience, and not fundamental to our results. Indeed, agents in real-world multi-agent systems may exchange messages spanning multiple tokens. Here, we explain how our analysis carries over to the setup where agents communicate multiple tokens in a message.

**Formalization in the case of multi-token messages**   Assume an agent $T_i$ aims to send a multi-token message $\sigma_1 \ldots \sigma_k$ (each token in $\Xi_{|x|}$) to agent $T_j$ The simplest way of formalizing multi-token messages is to simply view these as multiple single-token messages: $T_i$ iteratively communicates $\sigma_1, \ldots, \sigma_k$ over a total of $k$ time steps. We believe this to be natural and sensible approach, as transformer-based agents can naturally produce and consume one token per time step.

If a clear delineation of which sequences of single message tokens count as a multi-token message is desired, then a natural formalization is for the sending agent to output "START_MESSAGE" and "END_MESSAGE" tokens before and after the message tokens.

**Implications for Results**   Here, we discuss the implications for our theoretical results when messages span multiple tokens. By the above discussion, we can equivalently translate a protocol using multi-token messages into a protocol with single-token messages, without change to the size, width, computation depth, or number of communication tokens. Thus, bounds on size, width, and computation depth remain entirely unaffected by generalization to multi-token messages. The *communication budget*, i.e., the *number of message tokens*, also remains unaffected. As a consequence, all of our theoretical results remain unaffected even if the messages span multiple tokens.

## H   COMPARISON TO SELF-CONSISTENCY AND VOTING APPROACHES

As stated in the Introduction, our results concern the expressivity of multi-agent systems where each agent has access to a disjoint part of the input. Thus, the popular strategy of giving the same input to multiple agents and performing majority voting on their answers (e.g. Wang et al., 2022) is not in scope. Here, we show that multi-agent systems in our framework can achieve substantially better tradeoffs than majority voting-based approaches. Our argument is conceptually related to that in Mirtaheri et al. (2025), who however only compared voting (without any CoT) to long CoTs (in a single agent, without any communication), and relied on unproven conjectures about $\mathrm{TC}^0$. We instead compare majority voting and a more sophisticated multi-agent system (PrefixSum, Section 4.3), when both are allowed the same computation depth.

We consider state tracking in the simple case of PARITY. We contrast PrefixSum to majority voting between agents, and in both cases allow a depth (number of sequentially produced tokens for each agent) of $\mathcal{O}(\log N)$. We first give a general definition of majority voting in our framework. To make our result as strong as possible, and cover even voting among inhomogenous agents, we do not assume that the agents share parameters, but even allow voting among agents given by different models.

**Definition H.1** (Majority Voting)**.**  Consider $w(N)$ distinct agents, each given by UHAT transformers $T_1, \ldots, T_{w(N)}$. For full generality, we allow each agent to have their own parameters. We only assume that the number of layers and heads is uniformly bounded across agents. We run each agent on the full input, producing a CoT of length (computation depth) $depth(N)$, ending in a single-token response. We then output the most frequent response (with an arbitrary choice in the case of ties).

Despite the generality of this formalization, we now show a *super-polynomial* separation between majority voting and PrefixSum, by giving a lower bound on the number of agents needed to compute PARITY in a majority voting scheme. We consider the setup where the multi-agent system aims to compute PARITY state tracking exactly. This is consistent with our overall framework, where we expect that multi-agent systems perform tasks perfectly. Here, we show:

**Proposition H.1.** *Consider a majority voting scheme among $w(N)$ agents computing* PARITY *over length-$N$ inputs, with computation depth $depth(N) = \mathcal{O}(\log N)$.*

*This scheme must have $w(N) = 2^{\Omega(N^c)}$ agents, for some $c > 0$ depending on the number of heads and layers in the transformers representing the agents.*

The key takeaway here is the near-exponential lower bound on the number of agents, $w(N) = 2^{\Omega(N^c)}$. In contrast, PrefixSum attains perfect accuracy with computation depth $\mathcal{O}(\log N)$ and $w(N) = N$, as we showed in Prop. 4.5. This thus leads to a super-polynomial separation, between a linear number of agents in PrefixSum, and a near-exponential number of agents in majority voting.

We note that the above result concerns majority voting schemes that exactly compute PARITY with zero errors, based on results on voting polynomials by Aspnes et al. (1991). We conjecture that these techniques of can also be generalized to provide lower bounds on the number of agents needed for any voting scheme computing PARITY with good accuracy, though such a generalization is beyond our scope here. Importantly, given that PrefixSum solves PARITY deterministically, we believe that a bound on error-free computation is most relevant here.

*Proof.* Formally, we have $w(N)$ UHAT agents who all produce a CoT of length (communication depth) $depth(N) = \mathcal{O}(\log N)$. For a given string over the CoT alphabet, of length $depth(N)$, we can check in $AC^0$ that it is produced by a UHAT transformer. The number of layers of this circuit is bounded in terms of number of layers and heads of the transformer; the size is polynomial in $N$.

The number of such strings is $\mathcal{O}(poly(N))^{\mathcal{O}(\log N)} = \mathcal{O}(N^{C \log N})$. We now replicate this circuit once for every string, for each string checking if it is produced by the transformer. Note that the transformer produces exactly one of these. The replicate that detects that the string matches the CoT produced by the transformer signals this and outputs the result (0 or 1); all other replicates output 0. A single unbounded-fanin OR gate then extracts the answer from this one one replicate. We thus have obtained an AND-OR circuit computing the output of a single agent, with size $\mathcal{O}(N^{C \log N})$ for some $C \geq 1$, and with $d$ layers, for $d$ bounded across $N$.

In order to perform majority voting, we now put together all agents with a majority gate at the root. This is a circuit of size $s(N) = \mathcal{O}(w(N) \cdot N^{C \log N})$, with AND-OR gates and a single majority gate at the root providing the final output. Assume that this circuit correctly computes PARITY. By Lemma 5.4 in Aspnes et al. (1991), we have

$$s(N) = 2^{\Omega(N^{1/(4d)})} \tag{26}$$

We thus have

$$\log w(N) + \log N^{C \log N} = \Omega(N^{1/(4d)}) \tag{27}$$

or

$$\log w(N) + C \cdot (\log N)^2 = \Omega(N^{1/(4d)}) \tag{28}$$

which means

$$\log w(N) = \Omega(N^{1/(4d)} - (\log N)^2) \tag{29}$$

and hence

$$w(N) = 2^{\Omega(N^{1/(4d)})} \tag{30}$$

This proves the result.

$\square$

## I  IMPLICATIONS FOR FIXED PRECISION TRANSFORMERS

Our theoretical results are shown for UHAT, a popular theoretical model using hard attention. A reviewer asks about the implications of our results to soft attention transformers with finite precision and fixed width. The expressive power of causal transformers with soft attention, fixed precision, and fixed width was studied by Li & Cotterell (2025), who showed that their expressiveness corresponds to a subclass of the two-variable fragment of first-order logic over words, $\text{PFO}_2[<]$. This in turn can be simulated in UHAT, as shown in Jerad et al. (2025b). While they study the case of transformers without positional encoding (NoPE), expanding their results to absolute position encodings (APE) is straightforward:

**Proposition I.1.** *Let $T$ be a soft-attention transformer with finite precision and finite width, and absolute positional encodings $p_1, p_2, \dots \in \mathbb{R}^d$ with bounded norm, $\sup_i \|p_i\|_2 < \infty$. Then $T$ can be simulated by a UHAT transformer.*

*Proof.* The assumptions imply that the vectors $p_1, p_2, \ldots$ traverse a finite set $\mathcal{A}$, which we can bijectively map to the set $\{1, \ldots, |\mathcal{A}|\}$. Now to simulate the operation of $T$ on a string, we can view $T$ is a NoPE transformer operating over a string where each position $i$ is annotated with the index corresponding to $p_i \in \mathcal{A}$. As this NoPE transformer can be translated to UHAT, we can then augment the resulting UHAT transformer with positional encodings to simulate the operations of the original APE transformer $T$. □

As a consequence, all lower bounds on UHAT-based multi-agent systems in this paper transfer to such finite-precision softmax transformers. On the other hand, our experimental results in Section 5 show that protocols attaining asymptotically optimal tradeoffs can indeed be implemented using real transformers.

