# OpenReview forum: "Benefits and Limitations of Communication in Multi-Agent Reasoning"
_ICLR.cc/2026/Conference — ICLR 2026 Poster_

### Official Review · Reviewer_7TtA · 2025-10-30

**Soundness:** 3
**Presentation:** 3
**Contribution:** 3
**Rating:** 6
**Confidence:** 3

**Summary:**

This paper studies the trade-off between expressive power and communication cost in multi-agent reasoning systems.

**Strengths:**

1. The ternary structure of Size/Depth/Communication is used to characterize the "cost of scalability" in multi-agent reasoning, and an "impossible zone" is proved: when communication is O(1), it is impossible to reduce the depth to O(Size/w) as the number of agents increases (Proposition 4.2). This has significant implications for system design.

2. Section 3.1 and the discussion section map the three theoretical intervals to existing pipelines (such as CoA, LLM×MapReduce, LongAgent, hierarchical merging, etc.), making them operable.

**Weaknesses:**

1. The dataset is primarily synthetic and lacks specific comparisons on real-world document question answering, complex RAGs, or graph reasoning tasks.

2. The robustness to noise and instruction following failure is only explained in §5.2 as a phenomenon of "constant extra token overhead", lacking a systematic analysis.

3. Without integrating these protocols into actual multi-agent systems, it is impossible to observe their true guidance for MAS systems.

**Questions:**

1. How do the three types of protocols degrade under message loss/delay/content error? Can you provide an upper bound for noisy communication (e.g., converting O(k) rounds into a success probability with an error rate ε)?

2. Does the main proposition still maintain optimality under soft attention, finite precision, and fixed width?

3. Can you provide the token overhead required for training and inference in a specific scenario?

---

> ### Author Response · Authors · 2025-11-21
> **Official Comment by Authors**
>
> # Reviewer 7TtA
> We thank the reviewer for their constructive feedback. We believe this has led to an improved submission
> ## Weaknesses:
> > The dataset is primarily synthetic and lacks specific comparisons on real-world document question answering, complex RAGs, or graph reasoning tasks.
>
> We are currently working on additional experiments on real world data that we are hoping to deliver soon. We will update the reviewers when these are available and added to the main manuscript.
>
> We note however, that the aim of the experimental section is to validate the theoretical claims we provide. By using synthetic tasks, we have full control over problem complexity. This allows us to conduct experiments which better delineate the theoretical predictions we make.
>
> > The robustness to noise and instruction following failure is only explained in §5.2 as a phenomenon of "constant extra token overhead", lacking a systematic analysis.
>
> We believe you refer to *"We note, however, a slight increase in computational depth for high levels of communication. This is due to poor instruction following; models add a constant token overhead by repeating the query and explaining the procedure, especially noticeable in high-communication regimes.*"
> We understand this was not clearly formulated. What we meant was: While our instructions specified the specific format of the agents' outputs, the LLMs didn't always adhere to this and produced more tokens than needed. This tended to lead to a slight increase in the number of tokens beyond what would be theoretically needed.
> As our paper is concerned with the expressivity of multi-agent systems, we consider a systematic study of failures of instructional following out of scope; importantly, our empirical results show that multi-agent systems implemented with existing LLMs can implement protocols attaining the scaling predicted by our bounds.
>
> > Without integrating these protocols into actual multi-agent systems, it is impossible to observe their true guidance for MAS systems.
>
> Our experiments (both on synthetic datasets and the new experiments on real-world datasets) indeed integrate the protocols into actual multi-agent systems using pretrained LLMs as agents and using prompting to assign agents their roles.
> The protocols we implement in for our experimental results correspond to the "rule-based" protocol type as defined in the survey conducted by Tran et al. [1].
> We thus respectfully argue that our experiments do represent multi-agent systems.
>
> [1] Tran, Khanh-Tung, Dung Dao, Minh-Duong Nguyen, Quoc-Viet Pham, Barry O'Sullivan, and Hoang D. Nguyen. "Multi-agent collaboration mechanisms: A survey of llms." arXiv preprint arXiv:2501.06322 (2025).
> ## Questions:
> > How do the three types of protocols degrade under message loss/delay/content error? Can you provide an upper bound for noisy communication (e.g., converting O(k) rounds into a success probability with an error rate ε)?
>
> This is an interesting question, but one deserving separate systematic study in future work. Our work focuses on the expressivity of multi-agent systems based on Transformer LLMs. Probabilistic study of error bounds is beyond the scope of our current analysis, but remains an interesting direction for future work. We will clarify this more explicitly in the paper.
>
> As a first step towards this future direction, we added an error analysis for PARITY in Appendix D. Given the probability of failure $\epsilon$ of individual agents, our new analysis provides an analytical formula for the error for the entire protocol.
>
>
> > Does the main proposition still maintain optimality under soft attention, finite precision, and fixed width?
>
> We are unsure which "main proposition" the reviewer is referring to.
> Concurrent work by Li & Cotterell [1] and Jerad et al [2] show that Transformers with soft attention, finite precision, and fixed width are at most as expressive as UHAT.
> Hence, our optimality results also provide bounds on that setup. We add a detailed discussion of this in Appendix I.
>
>
>
> [1] Jiaoda Li and Ryan Cotterell. Characterizing the expressivity of fixed-precision transformer language models. NeurIPS 2025.
>
> [2] Selim Jerad, Anej Svete, Jiaoda Li, and Ryan Cotterell. Unique hard attention: A tale of two sides. arXiv 2025.
>
> > Can you provide the token overhead required for training and inference in a specific scenario?
>
> We'd like to clarify that implementing our protocols requires no further training, as we simply rely on instruction following capabilities. Regarding the token overhead required for inference, the empirical measure of "Computation Depth" that can be found in Fig. 4b and Fig. 5c is computed by summing the average token usage across every round of communication in the protocol. We hope this clears up the reviewers confusion.

---

> > ### Author Response · Authors · 2025-11-25
> > **Official Comment by Authors: New Experimental Result**
> >
> > We are happy to announce the addition of a novel experiment to our paper! Based on your feedback about real-world document QA, we have added to Section 5.1 evaluation on the needle-in-a-haystack benchmark (https://github.com/gkamradt/LLMTest_NeedleInAHaystack/tree/main). This tasks consists in finding a "needle" (answer to some query) in a "haystack" (large corpus of text). This closely matches the Recall regime in our theoretical analysis.
> >
> > We hope this addresses your concerns about experimental validation. If this is indeed the case, we kindly ask you to consider increasing your score.
> >
> > Please see the general comment for more information.
> >
> > In summary, we believe that we have addressed all of your concerns. Please let us know if you have any remaining questions or concerns!

---

> > > ### Comment · Reviewer_7TtA · 2025-11-26
> > > **Reply of Rebuttal and Score Update**
> > >
> > > Given that the authors have effectively addressed my main concerns and strengthened the empirical validation, I have decided to raise my score.

---

### Official Review · Reviewer_6i2W · 2025-10-31

**Soundness:** 2
**Presentation:** 3
**Contribution:** 2
**Rating:** 4
**Confidence:** 4

**Summary:**

This paper presents a theoretical framework for analyzing the expressivity of multi-agent systems in LLM reasoning tasks. The authors group multi-agent reasoning approaches in three algorithmic families (associative recall, state tracking, and k-hop reasoning) and propose theoretical claims on the number of agents required, the agents' communication structure, and speedups that can be achieved with respect to a problem and context size. The evaluation on synthetic benchmarks using Llama models provides the experimental evidence to support the theoretical claims.

**Strengths:**

1. The formalization of multi-agent systems as directed acyclic graphs in Definition 3.1provides a rigorous foundation for analysis. The distinction between CoT edges and communication edges is well-defined.
2. Figure 1 effectively illustrates the three discussed communication protocols making it easier to understand the paper.
3. The experimental results demonstrate that theoretical statements are closely aligned with practical evaluation results.

**Weaknesses:**

1. The Definition 3.1 states that the agents can only send/receive a single token at each time step. This restriction does not reflect the related work in multi-agent systems, where agents typically are not constrained to communicate through a single-token representation.
2. While Appendix D provides prompts and hyperparameters, some choices lack justification. For example, why 8 agents are used for majority voting, why the specific chunk sizes were chosen?
3. The framework explicitly excludes multi-agent debate, voting mechanisms, and self-consistency approaches, which are popular in related work. While the authors justify this by focusing on expressivity rather than stochasticity-based improvements, this limits the framework's applicability.

**Questions:**

1. Justify the hyperparameter choices (see Weakness 2).

---

> ### Author Response · Authors · 2025-11-21
> **Official Comment by Authors**
>
> We thank the reviewer for the close reading and feedback, which helped improve our paper.
>
> ## Weaknesses:
> > The Definition 3.1 states that the agents can only send/receive a single token at each time step. This restriction does not reflect the related work in multi-agent systems, where agents typically are not constrained to communicate through a single-token representation.
>
> As mentioned in the footnote at line 161, our results still hold for multi-token messages. The choice of single-token communication is to simplify the mathematics for the formal proofs. We added in Appendix G an in-depth discussion and explanation of how our framework extends to the setting where messages may span multiple tokens.
>
> > While Appendix D provides prompts and hyperparameters, some choices lack justification. For example, why 8 agents are used for majority voting, why the specific chunk sizes were chosen?
>
> See answer to related question.
>
>
> > The framework explicitly excludes multi-agent debate, voting mechanisms, and self-consistency approaches, which are popular in related work. While the authors justify this by focusing on expressivity rather than stochasticity-based improvements, this limits the framework's applicability.
>
> Expressivity and stochasticity are two complementary parts of multi-agent system analysis. In expressivity, we study an errorless model of computation to better understand the asymptotics regarding communication. Thus, studying stochasticity-based improvements is out of scope for our current work. That being said, we include two novel theoretical results which i) shed light on the separation between self-consistency/majority voting and PrefixSum in the case of state tracking, ii) extend PrefixSum to the stochastic imperfect regime. We explain the breadth of these results below:
>
> First, we added a new theoretical result where we formally prove that a **majority voting approach to State Tracking would require far more agents (near exponential in the input length) than the prefix sum-based protocol**, when keeping the number of CoT steps per agent comparable. This illustrates that popular voting and self-consistency approaches can be far less efficient compared to the protocols studied in this paper; this also agrees with out experimental results, where Majority Voting underperforms our protocols. This proof of this result as well as an extended discussion of its implications can be found in Appendix H.
>
> Second, we also added a theoretical analysis of error propagation in the PrefixSum protocol in the case of PARITY. Given an agent with a probability of failure $\epsilon$, we provide a formula which gives the error for the entire protocol
>
> ## Questions:
> > Justify the hyperparameter choices (see Weakness 2).
> * The number of agents and chunk size was chosen through a round of hyperparameter selection. We add these details to Appendix D. We succinctly answer here as well:
>     * For nb agents, we ablated over [2, 4, 8, 16] across all considered tasks and found that past 8 agents, there was no significant improvement in any of the tasks.
>     * For chunk size, we performed ablations as described below. Note that, given the chunk size changes with length, the chosen intervals are not the same across tasks.
>         * For Recall, we ablated over powers of two between 8 and 64 for each sequence length
>         * For Parity, we ablated over [2, 4, 8, 16] and found that best performance was attained with chunk size 8.
>         * For Permutations, we ablated [2,4] for both PrefixSum and Chain of Agents. Best performance was attained with chunk size 2.
>         * For k-hop reasoning, we ablated over chunk sizes [10, 20, 50] and found best performance at 20 across all three fact counts considered across all three fact counts (100, 200, 500) considered.

---

> > ### Author Response · Authors · 2025-11-25
> > **Official Comment by Authors**
> >
> > Thank you again for your helpful feedback!
> >
> > We believe that we have addressed in an actionable manner both of the main weaknesses you have raised. To summarize, we have
> > * Added Appendix Section G, which details how our formalization can be applied to the multi-token case.
> > * Added all necessary hyperparameter details for the experiments to the appendix.
> > * Added a novel theoretical result contrasting voting mechanisms to the provably optimal protocol (PrefixSum) in the case of state tracking.
> >
> > Let us know if you have any remaining questions or concerns! If you believe we have addressed your main concerns, we would appreciate you considering increasing your score.

---

### Official Review · Reviewer_YPeN · 2025-10-31

**Soundness:** 3
**Presentation:** 3
**Contribution:** 3
**Rating:** 6
**Confidence:** 3

**Summary:**

This paper aims to resolve: When and how does inter‑agent communication provably help LLM multi‑agent reasoning as problem size and context scale?  CoT and test‑time compute improve reasoning but degrade with long contexts; recent multi‑agent systems split tasks, yet their expressive capacity and resource tradeoffs lack theory. The work targets principled guidance for scalable designs by formalizing multi‑agent protocols as Transformer‑computable DAGs with size, computation depth (wall‑clock proxy), and communication budget, and asks for achievable/necessary tradeoffs across tasks.

**Strengths:**

Clear and actionable formalization (Section 3). The paper models multi-agent systems as Transformer-computable labeled DAGs and defines three concrete metrics: Depth(N), Size(N), and communication budget. Definitions 3.1–3.3 make the theoretical analysis both reproducible and extensible. This formalism is a valuable contribution to the emerging theory of LLM-based multi-agent systems. The approach seems inspired by work on GNN+MARL, such as https://arxiv.org/abs/2210.13148.

Proposition 4.1 (Conservation of size) and Proposition 4.2 (bounded communication implies O(1) single-agent CoT) together identify three feasible operating regimes. Importantly, they rule out the appealing but impossible scenario of "O(1) communication with strong depth reduction." The accompanying figure and discussion in Section 4.1 and Figure 2 are clear and immediately applicable to system design.

Protocol designs with matching bounds. The paper demonstrates that O(1) depth and communication is achievable (Proposition 4.3), providing formal justification for simple manager-worker designs on retrieval tasks (page 6). For more complex tasks, prefix-sum-style cascades achieve depth O(log w + N/w) with optimal communication Θ(w) and size Θ(N). The lower bounds in Proposition 4.7 match these upper bounds up to logarithmic factors (pages 6–7).

The experiments mirror the theoretical predictions well, which strengthens confidence in the framework's soundness. Figures 3–5 and the additional results in Appendix E (Figures 6–9) all align with the theoretical characterizations.

**Weaknesses:**

Idealized communication model. The theoretical model assumes single-token messages and uses a UHAT-style assumption. The proofs in Appendix C.2 rely on these idealizations to obtain the O(1) depth equivalences. It would strengthen the work to discuss robustness when messages span multiple tokens and agents don't perfectly adhere to the protocol. How much do the bounds degrade in more realistic settings?

Depth lower bounds may underestimate practical costs. The authors acknowledge that instruction-following overhead (extra boilerplate tokens) affects depth lower bounds in practice (Section 5.2). However, the cost model doesn't explicitly price this overhead in terms of compute or latency units, which makes it harder to apply the theoretical predictions to real systems.

Limited experimental scope. The experiments focus on synthetic tasks (recall, parity, k-hop reasoning) using Llama-3.* and EXAONE via TogetherAI. While these validate the theory, there's no evaluation on real-world long-context reasoning tasks like complex mathematical problem-solving, program synthesis, or long-document question answering. In these settings, memory management, tool use, and noisy communication patterns would provide a more complete picture of practical applicability (Section 5 & Appendix D).

Gap between theoretical metrics and practical deployment. While Definitions 3.1–3.3 are mathematically precise, an explicit mapping from "edges/tokens" to actual prompt token budgets and per-round latency would help practitioners. For example, when reading Figure 6, it would be valuable to understand how to translate the theoretical metrics into wall-clock predictions for system design.

**Questions:**

I'd like to discuss a few points with the authors:

Multi-token messages. Your proofs assume single-token communication, and you mention that extension to bounded-length words is "straightforward" (Section 3). Could you clarify how the depth and communication lower bounds would change when messages are b-token words? Specifically, how do the results differ when b is constant versus when b = O(log N)?

Robustness to asynchrony and failures. What happens if workers reply at different times or occasionally drop messages? Can the Conservation-of-size argument and Proposition 4.2 be recovered under such conditions, or does this break the O(1) single-agent simulation result? (Related to Appendix C.2)

Practical cost model. The figures show "computation depth" versus "communication edges," which are clean theoretical metrics. Could you provide a cost model that maps these to concrete latency (in terms of round-trips) and FLOPs? This would help designers choose w(N) under realistic compute budgets when implementing these protocols. (Related to Figure 6 and Appendix E)

---

> ### Author Response · Authors · 2025-11-21
> **Official Comment by Authors (1/2)**
>
> Thank you for taking the time to read our paper and giving a constructive review. We believe your comments helped improve the paper. We address the weaknesses and questions you raise point-by-point.
> ## Weaknesses:
> > Idealized communication model. The theoretical model assumes single-token messages and uses a UHAT-style assumption. The proofs in Appendix C.2 rely on these idealizations to obtain the O(1) depth equivalences. It would strengthen the work to discuss robustness when messages span multiple tokens and agents don't perfectly adhere to the protocol. How much do the bounds degrade in more realistic settings?
>
> We answer point-by-point:
> * Our formalization can be straightforwardly extended to the setting where messages span multiple tokens, and our results remain fully valid in that setup. We mainly consider single token communication to simplify proofs. We added an in-depth discussion of this in Appendix G.
> * UHAT is a common assumption made in the expressivity literature. Recent work by Li & Cotterell [1] and Jerad et al [2] show that Transformers with soft attention, finite precision, and fixed width are at most as expressive as UHAT. Hence, our optimality results also provide bounds in that setting. We added a detailed discussion of this topic in Appendix I.
> * We added an analysis of PARITY under the model of an imperfect agent in Appendix D as well as to Section 5.2.
>
> We hope the following additions clear up any misunderstandings concerning the formalization.
>
> [1] Jiaoda Li and Ryan Cotterell. Characterizing the expressivity of fixed-precision transformer language models. NeurIPS 2025.
>
> [2] Selim Jerad, Anej Svete, Jiaoda Li, and Ryan Cotterell. Unique hard attention: A tale of two sides. arXiv 2025.
> > Depth lower bounds may underestimate practical costs. The authors acknowledge that instruction-following overhead (extra boilerplate tokens) affects depth lower bounds in practice (Section 5.2). However, the cost model doesn't explicitly price this overhead in terms of compute or latency units, which makes it harder to apply the theoretical predictions to real systems.
>
> The cost of instruction following overhead is O(1), as it corresponds to the model repeating the instructions or the query. Thus, this does not meaningfully impact the predictive power of our theory w.r.t. real systems. We believe the main trends predicted by our theorertical results are still well reflected in our experimental validation despite this overhead.
>
> > Limited experimental scope. The experiments focus on synthetic tasks (recall, parity, k-hop reasoning) using Llama-3.* and EXAONE via TogetherAI. While these validate the theory, there's no evaluation on real-world long-context reasoning tasks like complex mathematical problem-solving, program synthesis, or long-document question answering. In these settings, memory management, tool use, and noisy communication patterns would provide a more complete picture of practical applicability (Section 5 & Appendix D).
>
> We are currently working on additional experiments on real world data that we are hoping to deliver soon. We will update the reviewers when these are available and added to the main manuscript.
>
> We note, however, that the aim of the experimental section is to validate the theoretical claims we provide. By using synthetic tasks, we have full control over problem complexity. This allows us to conduct experiments which better delineate the theoretical predictions we make.
>
> > Gap between theoretical metrics and practical deployment. While Definitions 3.1–3.3 are mathematically precise, an explicit mapping from "edges/tokens" to actual prompt token budgets and per-round latency would help practitioners. For example, when reading Figure 6, it would be valuable to understand how to translate the theoretical metrics into wall-clock predictions for system design.
>
> In Figures 4b and 6, we indeed use a calculation based on token usage metrics as a proxy for computation depth: we take the average usage at each round and take the sum across rounds to obtain our computation depth metric. The main factors which affect wall-clock time in our protocol design are the size of the input prompt as well as the number of CoT steps performed by an LLM agent before outputting its response. The prompt would increase by a constant offset the quadratic cost of attention. The number of CoT steps scales almost linearly in the number of steps.
>
> Wall-clock time would be easy to obtain by a similar computation as the token metrics; one could take the average wall-clock time for generation at each round and then sum to have the "average wall-clock time" for the protocol. Unfortunately, we did not log these quantities and thus cannot report these metrics.

---

> > ### Author Response · Authors · 2025-11-21
> > **Official Comment by Authors (2/2)**
> >
> > ## Questions:
> > I'd like to discuss a few points with the authors:
> >
> > > Multi-token messages. Your proofs assume single-token communication, and you mention that extension to bounded-length words is "straightforward" (Section 3). Could you clarify how the depth and communication lower bounds would change when messages are b-token words? Specifically, how do the results differ when b is constant versus when b = O(log N)?
> >
> > The choice of single token communication is to simplify the mathematics for the formal proofs.
> > As we mentioned in line 161, extension to multi-token messages is straightforward: Assume an agent $T_i$ aims to send a multi-token message $\sigma_1 \dots \sigma_k$ to agent $T_j$. Formally, this means that $T_i$ iteratively communicates $\sigma_1, \dots, \sigma_k$ over a total of $k$ time steps. Crucially, this can be equivalently simulated with a sequence of $k$ single-token messages. The size, width, computation depth, and number of communication tokens (communication budget) all remain unaffected in this equivalence. Hence, all of our results on these quantities remain transfer to the case where messages are b-token words, and are in fact invariant to $b$ constant or even $b = O(\log N)$.
> > An in-depth discussion of multi-token communication can be found in Appendix Section G.
> >
> > > Robustness to asynchrony and failures. What happens if workers reply at different times or occasionally drop messages? Can the Conservation-of-size argument and Proposition 4.2 be recovered under such conditions, or does this break the O(1) single-agent simulation result? (Related to Appendix C.2)
> >
> > The conservation of size result is an optimistic upper bound. The key idea of this proposition is that even within an idealized communication regime, no multi-agent protocol can reduce the total "size" or quantity of compute necessary to solve a task. Thus the result would remain true even under such degradations.
> >
> > > Practical cost model. The figures show "computation depth" versus "communication edges," which are clean theoretical metrics. Could you provide a cost model that maps these to concrete latency (in terms of round-trips) and FLOPs? This would help designers choose w(N) under realistic compute budgets when implementing these protocols. (Related to Figure 6 and Appendix E)
> >
> > See response to Weakness 3 (the last one)

---

> > > ### Author Response · Authors · 2025-11-25
> > > **Official Comment by Authors: New Experimental Result**
> > >
> > > We are happy to announce the addition of a novel experiment to our paper! Based on your feedback about real-world long context tasks, we have added to Section 5.1 evaluation on the needle-in-a-haystack benchmark (https://github.com/gkamradt/LLMTest_NeedleInAHaystack/tree/main). This tasks consists in finding a "needle" (answer to some query) in a "haystack" (large corpus of text). This closely matches the Recall regime in our theoretical analysis.
> > >
> > > We hope this addresses your concerns about experimental validation. If this is indeed the case, we kindly ask you to consider increasing your score.
> > >
> > > Please see the general comment for more information.
> > >
> > > In summary, we believe that we have addressed all of your concerns. Please let us know if you have any remaining questions or concerns!

---

> ### Comment · Reviewer_YPeN · 2025-11-28
>
> I thank the authors for the detailed response and the new Needle-in-a-Haystack experiment. The clarifications in Appendix G regarding multi-token message equivalence also resolve my theoretical concerns. While I still believe that evaluations on complex reasoning tasks (like math or code synthesis) would have provided a more complete picture of the "practical cost" of coordination (where instruction overhead and error propagation are more critical than in retrieval), I acknowledge that the current experiments sufficiently support the core theoretical claims. I am maintaining my positive score.

---

### Official Review · Reviewer_PxAy · 2025-11-04

**Soundness:** 3
**Presentation:** 3
**Contribution:** 3
**Rating:** 6
**Confidence:** 2

**Summary:**

The paper develops a formal framework to study multi-agent reasoning in LLMs, defining clear complexity measures such as width, depth, and communication. It analyzes three representative task families—associative recall, state tracking, and k-hop reasoning—and characterizes their optimal trade-offs. Both upper and lower bounds are derived, showing when communication helps and when it saturates. Empirical simulations with pretrained LLMs on synthetic tasks validate the theoretical predictions and illustrate the regimes where multi-agent protocols are beneficial.

**Strengths:**

- The paper lays out a clean formal model for multi-agent reasoning in LLM settings, making the resource trade-offs (width/depth/comm) precise.
-It analyzes three task families with sharp trade-offs, and the table does a good job of clearly summarizing the regimes.
- Both upper and lower bounds are provided in addition to constructions.

**Weaknesses:**

The paper does not include simulations on realistic tasks such as multi-document summarization or noisy graph question-answering, so it remains unclear how well the proposed framework extends to complex, real-world reasoning settings.

**Questions:**

- Have you considered running simulations on more realistic tasks—such as multi-document summarization or noisy graph QA—to test whether the theoretical trade-offs observed in your framework persist in practical, large-scale reasoning settings?
- How would your theoretical bounds change (if at all) when agents are allowed to overlap input chunks (i.e., share some context) or when messages can be multi-token (not just one token)? Have you thought about generalising to that regime, and do you expect qualitatively new regimes?

---

> ### Author Response · Authors · 2025-11-21
> **Official Comment by Authors**
>
> Thank you for taking the time to read our work and providing a constructive review. We respond point-by-point to the questions and weaknesses you raise.
> ## Weaknesses:
> > The paper does not include simulations on realistic tasks such as multi-document summarization or noisy graph question-answering, so it remains unclear how well the proposed framework extends to complex, real-world reasoning settings.
>
> See response to Question 1.
>
> ## Questions:
> > Have you considered running simulations on more realistic tasks—such as multi-document summarization or noisy graph QA—to test whether the theoretical trade-offs observed in your framework persist in practical, large-scale reasoning settings?
>
> We are currently working on additional experiments on real world data that we are hoping to deliver soon. We will update the reviewers when these are available and added to the main manuscript.
>
> We note, however, that the main aim of the experimental section is to validate the theoretical claims we provide. Using synthetic tasks allows us to have full control over problem complexity and  perform experiments which better delineate the theoretical predictions we make.
>
> > How would your theoretical bounds change (if at all) when agents are allowed to overlap input chunks (i.e., share some context) or when messages can be multi-token (not just one token)? Have you thought about generalising to that regime, and do you expect qualitatively new regimes?
> * For the algorithmic tasks we consider, overlap would mostly create redundancy in the reasoning performed by the agents. We answer by task:
>     * For recall/khop reasoning, this would mean multiple agents would process a same key-value pair. This would not change much to the theoretical results but, however, could be an interesting approach to increase performance in practical applications. By having more than one agent process a same subchunk of the context, we increase redundancy and thus the probability that possibly relevant information in a given subchunk is found/used by the system.
>     * For state tracking, this would not be feasible as the approach is based on the compositionality of nonoverlapping chunks; the result (e.g. the parity) of two overlapping chunks cannot be composed to produce the parity of the concatenation of those chunks.
>
> * Our formalization can be straightforwardly extended to multi-token communication. We focused on single token communication as it is simpler for the proofs. We have added an in-depth discussion of this in Appendix G.

---

> > ### Author Response · Authors · 2025-11-25
> > **Official Comment by Authors: New Experimental Result**
> >
> > We are happy to announce the addition of a novel experiment to our paper! Based on your feedback about realistic tasks, we have added to Section 5.1 evaluation on the needle-in-a-haystack benchmark (https://github.com/gkamradt/LLMTest_NeedleInAHaystack/tree/main). This tasks consists in finding a "needle" (answer to some query) in a "haystack" (large corpus of text). This closely matches the Recall regime in our theoretical analysis.
> >
> > We hope this addresses your concerns about experimental validation. If this is indeed the case, we kindly ask you to consider increasing your score.
> >
> > Please see the general comment for more information.
> >
> > In summary, we believe that we have addressed all of your concerns. Please let us know if you have any remaining questions or concerns!

---

> > > ### Comment · Reviewer_PxAy · 2025-11-26
> > >
> > > I acknowledge that the rebuttal addresses all of my concerns. I updated my score accordingly.

---

### Author Response · Authors · 2025-11-21
**General Response**

We thank the reviewers for their valuable feedback. We highlight the following strengths raised by the reviewers:

* **Clarity and rigour of the formalization.** Many reviewers mentioned that the graph formalization of a multi-agent protocol was clean and provided a rigorous foundation to the paper. Moreover, many reviewers highlighted the exposition of theoretical results as well, stating that Figure 1 as well as Table 1 made the theoretical contributions easy to understand.
* **Practical relevance of the theoretical results.** Many reviewers also pointed out the results were actionable, relevant to practice and had significant implications for system design.
* **Matching bounds for all constructions.** Another theoretical strength highlighted by a majority of reviewers was the proofs of optimality for the protocols presented in the work. These proofs provide strong guarantees about the considered protocols.


Based on the feedback given by the reviewers, we add the following to our manuscript:
* A novel theoretical result which shows a super-polynomial separation between majority voting and PrefixSum for the PARITY task. A detailed discussion of the result as well as a proof of the statement can be found in Appendix H.
* A section in Appendix I which discusses the relationship between our current model and models with finite precision/softmax attention.
* A discussion in Appendix Section G which discusses extensions to the multi-token communication case.
* A novel theoretical analysis of error propagation in the PrefixSum protocol for the PARITY task. We give a closed form formula for the probability of failure of the protocol given the failure rate of a single agent. We back up this theoretical result with experiments estimating protocol failure rate given empirical data for the failure rate of a single agent. The analysis as well as the proofs can be found in Appendix D.
* Note that we are also working on **additional experiments on real world data** that we are hoping to deliver soon.

We hope that these additions to the paper address the  questions raised by the reviewers. We believe these modifications lead to an improved submission.

---

> ### Author Response · Authors · 2025-11-25
> **Update: New Experimental Result**
>
> Some Reviewers (YPeN, PxAy, 7TtA) note that experiments in our original submission used synthetic data.
>
> In light of their comments (YPeN, PxAy, 7TtA), we have added an experiment on real world data to the main text. We use the needle-in-a-haystack benchmark (https://github.com/gkamradt/LLMTest_NeedleInAHaystack/tree/main) to evaluate the ability to perform recall. This tasks consists in finding a "needle" (answer to some query) in a "haystack" (large corpus of text). This task closely matches the Recall regime in our theoretical analysis.
>
> We test both the MajorityVoting baseline and the ChainOfAgents protocol (which we prove optimal for this task).  Importantly, the experimental results closely agree with our theoretical results: the token overhead across chunk size and context lengths remains basically constant. We also find that the provably optimal approach according to our theoretical results (i.e., ChainOfAgents) yields better performance across context length and needle depth than MajorityVoting.
>
> We add these novel results in Section 5.1 of the main paper. Experimental details for this can be found in Appendix Section E.5. We thank again the reviewers for their feedback. We believe the addition of this experiment has improved the quality of our submission.
>
> We recall that the primary contribution of our work is to better understand the asymptotics of multi-agent computation/communication. This is chiefly achieved by our theoretical analysis, but the choice of synthetic tasks is also crucial; this lets us systematically ablate over different problem parameters to observe the trends we predict in theory.

---

### Author Response · Authors · 2025-12-01
**Official Comment by Authors**

Dear AC and Reviewers,

We thank you for your time and valuable feedback. Given the current circumstances with OpenReview, we thought it could be insightful to summarize the current state of the rebuttal.

We initially received scores 6, 6, 6, 4. Based on reviewer feedback, we added the following to our manuscript:
* A novel theoretical result showing a super-polynomial separation between majority voting and PrefixSum for the PARITY task.
* A section in Appendix I discussing the relationship between our current model and models with finite precision/softmax attention.
* A discussion in Appendix Section G which discusses extensions to the multi-token communication case.
* A novel theoretical analysis of error propagation in the PrefixSum protocol for the PARITY task in Appendix Section D.
* A novel experiment on the needle-in-a-haystack benchmark (https://github.com/gkamradt/LLMTest_NeedleInAHaystack/tree/main) corroborating our theoretical results for the recall task family.

Following these additions, two reviewers increased their score from 6 to 8 and one decided to maintain their positive score of 6. Reviewer 6i2W, unfortunately, did not respond to our rebuttal before the decision to freeze the review process.

We are extremely grateful for the feedback given by the reviewers, and sincerely believe their recommendations improved the paper. We are happy to answer any question the new AC might have about the paper or the rebuttal.

Best wishes,

The Authors

---

### Meta-Review · Area_Chair_hzUw · 2025-12-26

**Summary:**

This paper presents a theoretical framework for analyzing the expressivity of a Transformer-based multi-agent system. From three types of tasks (associative recall, state tracking, and k-hop reasoning), this paper derives bounds on the number of agents, communication quality/structure, and speedups for three tasks. These theoretical results highlight the trade-off in the condition that communication benefits the scalability. This paper also provides empirical results on synthetic datasets with LLMs to support the derived theoretical analysis.

The initial score of this paper is 6,6,6,4. Reviewers summarized the paper's strengths, including its clear presentation, rigorous theoretical analysis, and contributions to LLM-based multi-agent systems, as well as the provision of principal insights. Concerns from reviewers primarily focused on the experimental scope, which mainly involved synthetic datasets, the applicability of conclusions to practical tasks, the configuration of hyperparameters, and the extension to more complex settings.

In rebuttal, the authors addressed these concerns by incorporating new theoretical results (e.g., super-polynomial separation for majority voting and PrefixSum in the PARTY task, error propagation analysis for PrefixSum), new discussion (e.g., Finite-precision models, softmax attention, and multi-token extensions), and new empirical results (e.g., needle-in-a-haystack problem). These supplementary results and clarification enhance the solidity of this paper and extend the applicability scope. During the rebuttal, two reviewers with a 6 score (PxAy, 7TtA) clearly expressed their willingness to raise the score. While the reviewer of 4 score (6i2W) did not raise the score, this reviewer has not responded to the authors' rebuttal with new questions or concerns.

Given the strong theoretical contribution to multi-agent systems, the active behavior of the authors in rebuttal and paper revision, and positive responses from the reviewers, I recommend acceptance as a poster.

**Reviewer Concerns:**

Reviewers’ concerns mainly include that the evaluated benchmarks are synthetic without real-world validations, Idealizations in the theoretical framework and bounds, generalization and extensions of the theory, and limited experimental scope. In rebuttal, the authors addressed all concerns from all four reviewers. Three reviewers (PxAy, 7TtA, YPeN) clearly express that their concerns are well-solved. While reviewer 6i2W did not respond to the rebuttal of the authors, the authors still provide a positive response in theory that covers all concerns.

**Reviewer Scores:**

In rebuttal, three reviewers (PxAy, 7TtA, YPeN) have provided the final decision of their scores that they are satisfied with the author’s rebuttal, where two of the reviewers (PxAy, 7TtA) have raised their score, and one of the reviewers (YPeN) has maintained the initial score.

On the other hand, one reviewer (6i2W) did not participate fully in this discussion and maintained the score of 4. Given that the authors thoroughly addressed all concerns raised by this reviewer, it is reasonable to expect that the score would likely remain unchanged or potentially increase in the absence of additional questions or objections.

---

### Decision · Program_Chairs · 2026-01-26

Accept (Poster)